# Enhancement of phase transition temperature through hydrogen bond modification in molecular ferroelectrics

Yu-An Xiong[1,3], Sheng-Shun Duan[2,3], Hui-Hui Hu[1,3], Jie Yao[1], Qiang Pan[1], Tai-Ting Sha[1], Xiao Wei[2], Hao-Ran Ji[1], Jun Wu [2] ✉ & Yu-Meng You [1] ✉

Molecular ferroelectrics are attracting great interest due to their light weight, mechanical flexibility, low cost, ease of processing and environmental friendliness. These advantages make molecular ferroelectrics viable alternatives or supplements to inorganic ceramics and polymer ferroelectrics. It is expected that molecular ferroelectrics with good performance can be fabricated, which in turns calls for effective chemical design strategies in crystal engineering. To achieve so, we propose a hydrogen bond modification method by introducing the hydroxyl group, and successfully boost the phase transition temperature ($T_c$) by at least 336 K. As a result, the molecular ferroelectric 1-hydroxy-3-adamantanammonium tetrafluoroborate [(HaaOH)BF$_4$] can maintain ferroelectricity until 528 K, a $T_c$ value much larger than that of BTO (390 K). Meanwhile, micro-domain patterns, in stable state for 2 years, can be directly written on the film of (HaaOH)BF$_4$. In this respect, hydrogen bond modification is a feasible and effective strategy for designing molecular ferroelectrics with high-$T_c$ and stable ferroelectric domains. Such an organic molecule with varied modification sites and the precise crystal engineering can provide an efficient route to enrich high-$T_c$ ferroelectrics with various physical properties.

Due to the unique feature of switchable spontaneous polarization, ferroelectric are widely used in industrial and commercial applications, such as ferroelectric random access memories, piezoelectric sonar, sensors, and electromechanical transformers[1–4]. A great majority of excellent and advanced ferroelectrics are based on inorganic ceramics and polymers, like BaTiO$_3$ (BTO), Pb(Zr,Ti)O$_3$ (PZT), polyvinylidene fluoride, etc.[5–9]. In recent years, molecular ferroelectrics emerge to be a focus of research, with the advantages of mechanical flexibility, lightweight, environmental friendliness, low cost, and ease of processing into films. Over a century has passed since the discovery of the molecular ferroelectric Rochelle salt. Recently, abundant progress has been made in molecular

ferroelectrics due to the convenient modification and design of crystal engineering[10–14]. The current research on molecular ferroelectrics is focused on new materials, methodological advancements, improving performance, etc. The practical applications of molecular ferroelectrics still need further exploration. Meanwhile, molecular ferroelectrics can serve as a complement to inorganic and polymeric materials in certain specialized applications. With continuous performance upgrade, molecular ferroelectrics can outperform inorganic and polymer ferroelectrics. For instance, diisopropylammonium bromid has the spontaneous polarization of 23 µC cm$^{-2}$ (close to that of BTO)[15]; 2-(hydroxymethyl)−2-nitro-1,3-propanediol obtains 48 crystallographically equivalent polarization

[1]Jiangsu Key Laboratory for Science and Applications of Molecular Ferroelectrics, Southeast University, Nanjing 211189, People's Republic of China. [2]Joint International Research Laboratory of Information Display and Visualization, School of Electronic Science and Engineering, Southeast University, Nanjing 210096, People's Republic of China. [3]These authors contributed equally: Yu-An Xiong, Sheng-Shun Duan, Hui-Hui Hu. ✉e-mail: wujunseu@seu.edu.cn; youyumeng@seu.edu.cn

directions, the largest amount among molecular ferroelectrics[16]; trimethylchloromethyl ammonium trichloromanganese (II) has a high piezoelectric coefficient ($d_{33}$ = 383 pC N$^{-1}$)[17,18]; and trimethyl-chloromethyl ammonium tetrachlorogallium(III) has a large piezo-electric voltage coefficient ($g_{33}$ = 1318 × 10$^{-3}$ Vm N$^{-1}$)[19]. In addition, molecular engineering, such as morphotropic phase boundaries[20], bandgap regulation[21,22], and ferroelectric domain engineering, such as vortex domain structures[23,24], strain-induced periodic domain structures[25] and design of charged domain walls[26], have been successfully carried out in molecular ferroelectrics. Based on these, researchers have introduced the quasi-spherical theory, introducing homochirality, and H/F substitution to summarize the molecular design principles of ferroelectrochemistry. In this sense, organic cations with more modification sites can be modified in a more diversified way. This has boosted the development of molecular ferroelectrics[12]. Consequently, it is promising that organic cations with more modification sites can facilitate the crystal engineering and performance optimization of molecular ferroelectrics.

As a significant intermolecular force, the hydrogen bond plays a key role in inducing ferroelectric polarization and promoting the phase transition temperature ($T_c$). As we all know, molecules based on the hydroxyl group, a fundamental contributor to inducing the hydrogen bond, have been synthesized in molecular ferroelectrics. Typical representatives include 2-(hydroxymethyl)-2-nitro-1,3-propanediol[16], $R/S$-3-quinuclidinol[27,28] and N-fluormethyltropine[29]. However, these molecules cannot maintain ferroelectricity at a high temperature, as the hydrogen bond network is not tight enough. Therefore, it is urgent to clarify and explore the feasibility of using hydrogen bonds to design molecular ferroelectrics effectively. The appropriate host and guest need to be selected to construct a hydrogen bond network of intermolecular interactions. This is a critical step to introduce ferroelectric polarization and improve $T_c$ values.

As a large spherical molecule, adamantane has modification sites of 10 C atoms, which has more C atoms and higher mass than its counterparts in molecular ferroelectrics, such as 1,4-diazabicyclo[2.2.2]octane, quinuclidine, and tropine. Here, based on the hydrogen bond modification strategy, 1-hydroxy-3-adamantanammonium (HaaOH$^+$) and BF$_4^-$ were chosen as the organic guest and host respectively to form a molecular ferroelectric (Supplementary Fig. 1). Also, by modifying the hydrogen bond of non-ferroelectric 1-adamantanammonium tetrafluoroborate [(Haa)BF$_4$] which has large-scale adamantane molecules, we successfully introduced polarization and enhanced $T_c$ by at least 336 K. To our knowledge, the temperature enhancement has reached a high level in molecular ferroelectrics (Supplementary Table 1). And the ferroelectricity was retained until the decomposition temperature of (HaaOH)BF$_4$ ($T_d$ = 528 K), which was closed to $T_c$ of [(2-aminoethyl)trimethylphosphanium]PbBr$_4$, a high $T_c$ in reported molecular ferroelectrics[30]. This further proved that hydrogen bond modification was an effective strategy in designing molecular ferro-electrics. The regular micron ferroelectric domain pattern was written on the thin film of (HaaOH)BF$_4$, a pattern in stable state for more than 2 years. Based on the stable polarization, the piezoelectric energy-harvesting device of (HaaOH)BF$_4$ has good efficient piezoelectric performance, which can light up 9 blue light-emitting diodes (LEDs) and has sensitive self-powered sensing. This will promote the application of molecular ferroelectrics in micro-nano electronic devices. Undoubtedly, the design strategy of hydrogen bond modification has great significance in optimizing the performance of molecular ferroelectrics, and the adamantane organic molecule with such a large number of modification sites will also provide more possibilities to develop molecular ferroelectrics.

## Results

The colorless and transparent bulk crystals of (HaaOH)BF$_4$ and (Haa)BF$_4$ were obtained by slowly evaporating the deionized aqueous

solutions of 1-hydroxy-3-adamantanamine and 1-adamantanamine in HBF$_4$ at room temperature, respectively. The bulk phase purity was verified by powder X-ray diffraction (PXRD) (Supplementary Fig. 2). Single-crystal X-ray diffraction analyses indicate that (Haa)BF$_4$ was crystallized in the orthorhombic central symmetric space group *Pnma* (point group mmm) at 293 K (Supplementary Table 2). It contains a Haa$^+$ organic cation and a BF$_4^-$ anion in a unit (Fig. 1b), and both the organic cation and anion are in partial disorder at room temperature. Hydroxyl modification was used to induce hydrogen bonds on the Haa$^+$ organic cations of the compound (Haa)BF$_4$. As shown in Fig. 1a, the designed organic cation HaaOH$^+$, as the template guest, was assembled with BF$_4^-$ into the hydrogen-bonded host-guest compound (HaaOH)BF$_4$. Single-crystal X-ray diffraction analyses indicate that (HaaOH)BF$_4$ was crystallized in the orthorhombic non-centrosymmetrical polar space group *Pna2$_1$* (point group *mm*2) at 293 K through hydrogen bond modification (Supplementary Table 2). This suggests that (HaaOH)BF$_4$ may be a ferroelectric. The crystal morphologies of (Haa)BF$_4$ and (HaaOH)BF$_4$ (Supplementary Fig. 2) are the same as that predicted by the Bravais, Friedel, Donnay, and Harker (BFDH) method (Supplementary Figs. 3 and 4). As denoted by the symmetry of the crystal point group, the polarization of (HaaOH)BF$_4$ is along the $c$-axis. Using the Berry phase method in VASP and based on the crystal structure, the simulated polarization was calculated to be 4.21 μC cm$^{-2}$ along the [001] direction at 293 K[31,32].

From the perspective of crystallographic stacking, only three N–H⋯F hydrogen bonds, which can be divided into two types, are formed on each Haa$^+$ organic cation. As depicted in Fig. 1d, guest-guest hydrogen bonds are absent between organic cations, and molecules are not fixed in an orderly state. However, in addition to the electro-static effect, the hydrogen bond of (HaaOH)BF$_4$ has a significant effect in inducing the polarization. There are four various types of hydrogen bonds between anions and cations in (HaaOH)BF$_4$ (Supplementary Table 3). And five hydrogen bonds are formed on a HaaOH$^+$ organic cation. In addition to the two N–H⋯F hydrogen bonds, there are also two N–H⋯O and one O–H⋯F hydrogen bonds. Molecules are orderly arranged under a stable hydrogen bond network (Fig. 1c). The crystal structure diagram shows that the hydroxy-modified organic cations are connected by N1–H1B⋯O1 to form guest-guest interactions (Supplementary Fig. 5). Besides, each organic cation connects to the same BF$_4^-$ anion through N1–H1C⋯F1B and O1–H1⋯F1A. Meanwhile, the three F atoms on each BF$_4^-$ anion connect with two HaaOH$^+$ organic cations through N1–H1A⋯F1C, N1–H1C⋯F1B, and O1–H1⋯F1A. The hydrogen bond network is constructed in (HaaOH)BF$_4$ through hydroxyl modification, resulting in the stable arrangement of ordered organic and inorganic ions.

We simulated and analyzed the organic cations of Haa$^+$ and HaaOH$^+$ through Hirshfeld surface analysis[33]. As for the two compounds, their anions are the same, while the organic cations are different. This difference is the main inducing factor of the varying crystal structures and phase transition behaviors. Then the electron density distributions around the HaaOH$^+$ (Fig. 2a) and Haa$^+$ (Fig. 2b) were analyzed respectively. Specifically, the hydrogen bond formed between the internal and external molecules of the Hirshfeld surface has strengthened the intermolecular force and reduced the distance between the donor and the receptor. Meanwhile, the calculated standard distance ($d_{norm}$) decreases and the $d_{norm}$ surface is displayed quite distinctly on the Hirshfeld surface in red (Fig. 2a, b). Similarly, hydrogen bonding has also shortened the atomic distance inside and outside the Hirshfeld surface ($d_e$ and $d_i$ are also of small values). In addition, $d_e$ and $d_i$ increases as the distance between the hydrogen bond and the point on the Hirshfeld surface increases. Thus, peaks are formed on the 2D fingerprint plots of (HaaOH)BF$_4$ (Fig. 2c) and (Haa)BF$_4$ (Fig. 2d). As shown in the decomposed fingerprint plots, the contribution proportions of the H⋯F contacts of

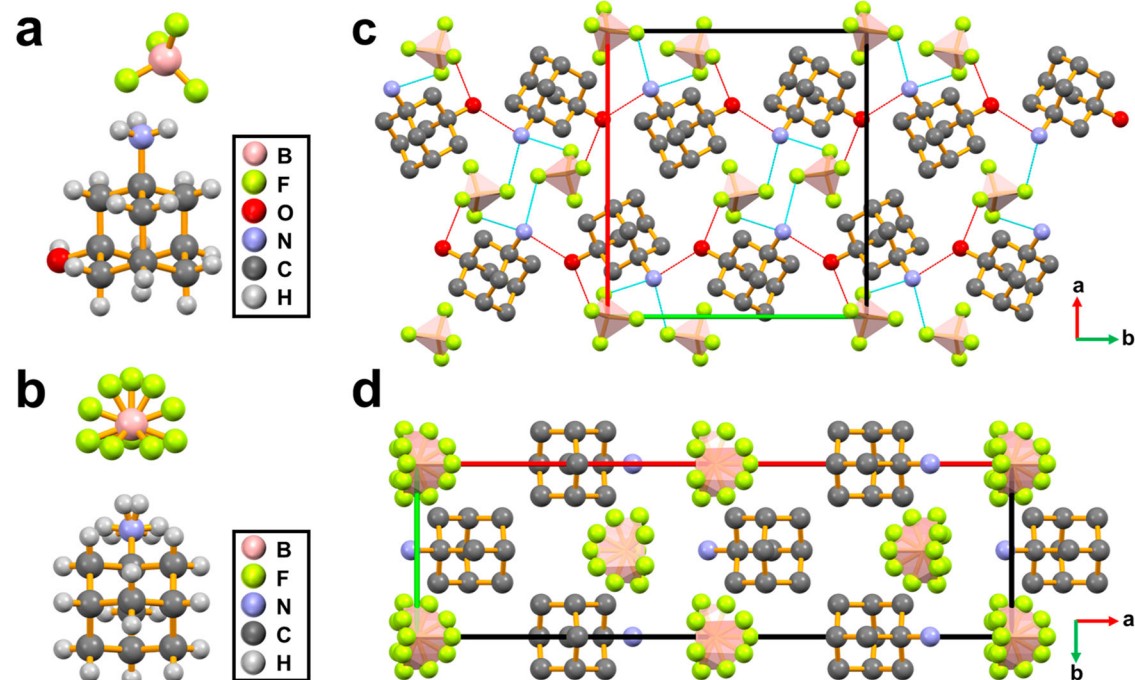

**Fig. 1 | Comparison of crystal structures between (HaaOH)BF₄ and (Haa)BF₄ at 293 K.** Asymmetric units of **a** (HaaOH)BF₄ and **b** (Haa)BF₄. **c** Packing view along the *c*-axis of (HaaOH)BF₄.The anions and cations are ordered. The dotted lines represent hydrogen bonding interactions. **d** Packing view along the *c*-axis of (Haa) BF₄. The anions and cations are orientationally disordered. Parts of hydrogen atoms are omitted for clarity.

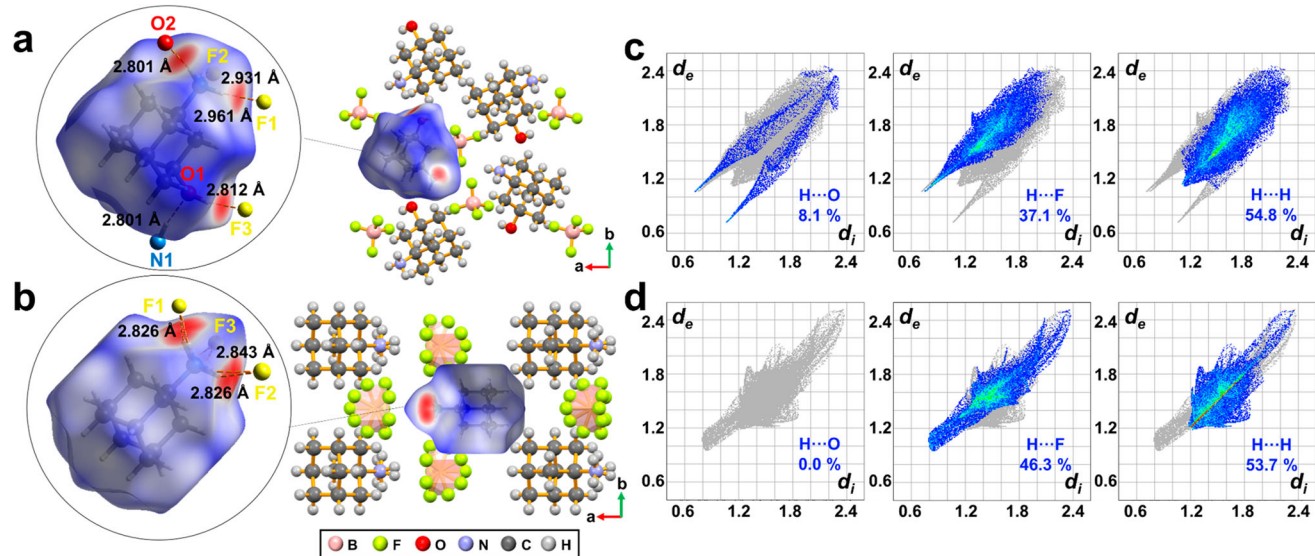

**Fig. 2 | Hirshfeld surface analyses of (HaaOH)BF₄ and (Haa)BF₄.** Hirshfeld surfaces of the guest cations in (HaaOH)BF₄ **a** and (Haa)BF₄ **b** whose $d_{norm}$ values are in the range of −0.6215 (red) to 1.4037 (blue) and −0.5902 (red) to 1.5298 (blue), respectively. The red spots represent the strong short-term contacts between neighboring atoms. Decomposed fingerprint plots and proportion for the H···O, H···F, and H···H contacts of the guest cations in (HaaOH)BF₄ **c** and (Haa)BF₄ **d** on the Hirshfeld surfaces are displayed respectively.

(HaaOH)BF₄ and (Haa)BF₄ in the Hirshfeld surface area were 37.1 % and 46.3 %, respectively. This is related to the host-guest interaction between organic cations and inorganic anions. Among all host-guest interactions in (HaaOH)BF₄, the strongest is the O−H···F hydrogen bond with $d_{O···F}$ of 2.812 Å. Two N−H···F hydrogen bonds with $d_{N···F}$ of 2.931 Å and 2.961 Å also play a major part (Fig. 2a). However, due to the disorder of H and F atoms, (Haa)BF₄ possesses more object-object interactions. As a result, the proportion of H···F in (HaaOH) BF₄ is slightly lower than that of (Haa)BF₄. But significant differences

still exist in the host-guest interaction between (HaaOH)BF₄ and (Haa)BF₄.

The adjacent organic cations in (HaaOH)BF₄ and (Haa)BF₄ have completely different guest-guest interactions. In the decomposed fingerprint plots shown in Fig. 2c, (HaaOH)BF₄ has a pair of symmetrical peaks, which is contributed to the N−H···O hydrogen bond in HaaOH⁺. Because the atoms forming the N−H···O hydrogen bond are symmetrically organized on the Hirshfeld surface based on internal molecules, the 2D fingerprint plot shows symmetric peaks

between $d_e$ and $d_i$ that have the same value. The guest-guest H···O interaction accounts for 8.1 % and 0 % in (HaaOH)BF$_4$, and (Haa)BF$_4$, respectively (Fig. 2c, d). And the $d_{N···O}$ of the N–H···O hydrogen bond is 2. 801 Å, the shortest among all hydrogen bonds in (HaaOH)BF$_4$. This implies that the attractive guest-guest interaction between adjacent organic cations is due to hydrogen bond modification. Furthermore, the intermolecular interactions were visualized quantitatively based on the energy framework analysis (Supplementary Fig. 6)[34,35]. The calculated energies between ions were represented by cylinders and are proportionate to the cylinders' thickness (Supplementary Fig. 7). As shown in the 3D topologies of the energy framework, ions were strongly connected through hydrogen bonds. These strong interactions form several stable columns, which would be difficult to destroy.

The chemical design method of hydrogen bond modification has formed stronger intermolecular interactions. This increases the energy barrier to the free rotation of HaaOH$^+$ cations and calls for a higher $T_c$ for the order-disorder transition of (HaaOH)BF$_4$. The differential scanning calorimetry (DSC) measurement shows that the $T_c$ of (Haa)BF$_4$ is 192 K (Supplementary Fig. 8). The temperature-dependent single-crystal X-ray diffraction shows that the low-temperature phase of (Haa)BF$_4$ crystallizes in the space group $P2_1/c$ (point group 2/$m$). And the distance between atoms and the intermolecular force have changed a little from the high-temperature phase due to the ordering of organic cations and anions and structural phase transition (Supplementary Fig. 9). To our amazement, the introduction of the hydroxyl group has changed the guest-guest and host-guest interactions of (HaaOH)BF$_4$ and formed a network of intermolecular forces (Supplementary Fig. 7). This makes (HaaOH)BF$_4$ should have a higher $T_c$. However, under the combined affection of hydrogen bonds and large HaaOH$^+$ cations, the potential $T_c$ is higher than the low decomposition temperature ($T_d$ = 528 K) (Supplementary Fig. 10). Meanwhile, the dielectric constant and loss in (HaaOH)BF$_4$ were probed by the temperature- and frequency-dependent dielectric permittivity measurements across various frequencies ranging from 500 Hz to 1 MHz (Supplementary Fig. 11). The real part ($\varepsilon'$) (Supplementary Fig. 11a) and the imaginary part ($\varepsilon''$) (Supplementary Fig. 11b) of the dielectric constant were found to gradually increase upon increasing the temperature and absent of structural phase transition. Thus, the crystallographic phase transition cannot be obtained before decomposition. (HaaOH)BF$_4$ experiences no crystal phase transitions and still maintains its ferroelectricity until $T_d$, a temperature close to the high $T_c$ = 534 K of reported molecular ferroelectrics[30]. More importantly, the $T_c$ enhancement (at least 336 K), as compared to $T_c$ of the parent compound (Haa)BF$_4$, is high among reported enhancements for molecular ferroelectrics. The enhancement is larger than the previous record of 288 K from [(4-methoxyanilinium)(18-crown-6)][BF$_4$] ($T_c$ = 127 K) to [(4-methoxyanilinium)(1-crown-6)][bis(trifluoromethanesulfonyl)ammonium] ($T_c$ = 415 K)[36]. This verifies that the hydrogen bond modification is effective in designing molecular ferroelectrics with a high $T_c$. Therefore, further exploration on the correlation between molecular structure and ferroelectricity is made possible.

Ferroelectric materials, possessing spontaneous polarization ($P_s$) with hysteresis effects, can be used for information storage, and the polarization of ferroelectric thin films can be switched at low voltages. Thin films of molecular ferroelectrics, represented by (HaaOH)BF$_4$, can be prepared at low temperature in a cost-effective and easy way. This makes these molecular ferroelectrics suitable for preparing ferroelectric electronic devices. By dropping the homogeneous deionized water solution of (HaaOH)BF$_4$ onto a fresh ozone-treated indium tin oxide (ITO)-coated conductive glass, the continuous block crystal film with high coverage was grown at the controlled temperature of 333 K. Then based on the measurements of the typical polarization−voltage ($P$–$V$) hysteresis loop, the

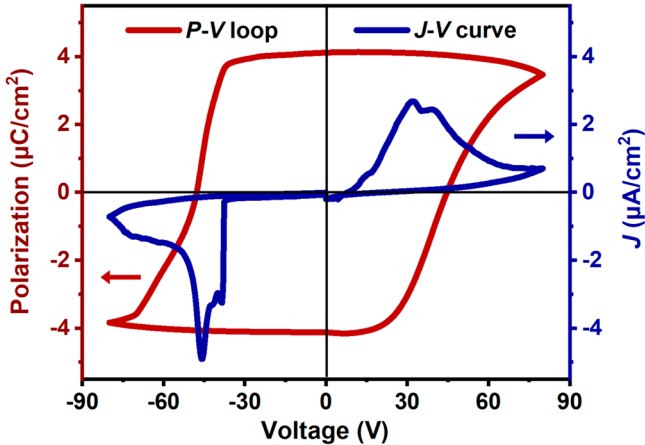

**Fig. 3 |** $J$–$V$ curve and $P$–$V$ hysteresis loop of the (HaaOH)BF$_4$ film at room temperature.

ferroelectricity of (HaaOH)BF$_4$ has been verified. The typical ferroelectric current density−voltage ($J$–$V$) curve and $P$–$V$ hysteresis loop were measured on the film of (HaaOH)BF$_4$ by the double-wave method at room temperature. This indicates that (HaaOH)BF$_4$ has obtained the reversible spontaneous polarization (Fig. 3). The typical ferroelectric $J$–$V$ curve shows two opposite current peaks. According to the $J$–$V$ curve, we obtained the perfect $P$–$V$ hysteresis loop by integrating the current. With an applied voltage, the $P_s$ value rapidly increases and reaches the maximum value of about 4.1 µC cm$^{-2}$ for thin film, which is close to the estimated value of 4.21 µC cm$^{-2}$ depending on the Berry phase method. The hysteresis loop is a significant feature of ferroelectric molecules, which proves (HaaOH)BF$_4$ as a ferroelectric. In this respect, the modification of hydrogen bond is an effective strategy to successfully introduce ferroelectric polarization.

Ferroelectricity characterization was carried out on the film of (HaaOH)BF$_4$ via the piezoresponse force microscopy (PFM) technology, realizing the polarization switching at micro and nano scales. We used the probe to scan across the surface of the thin film under the contact mode while applying an ac voltage to the ferroelectric samples simultaneously. Then polarization-dependent deformation occurred to the samples under the ac voltage due to the inverse piezoelectric effect. This can produce a nondestructive visualization of ferroelectric domains with ultra-high spatial resolution and help obtain the polarization information of these ferroelectric samples. PFM contains both lateral (LPFM) and vertical (VPFM) modes, corresponding to the in-plane and out-of-plane polarization components, respectively. According to the intensity and orientation of polarization components, the PFM amplitude and phase data can be obtained, respectively. Using the BFDH method, the crystal plane with dominant growth directions is predicted to be (110), and the polarization of (HaaOH)BF$_4$ is distributed in an in-plane way along the $c$-axis. Therefore, the domain structure of the (HaaOH)BF$_4$ film was detected by LPFM. Figure 4a, b show the as-grown domain structure of the (HaaOH)BF$_4$ film. The PFM phase imaging (Fig. 4b) shows a clear domain structure with a 180° contrast, due to the different polarization directions on both sides of the PFM probe. As depicted in Fig. 4a, the PFM amplitude imaging shows clear domain walls, that is, the boundary between two domains, which conform to the domain structure of PFM phase imaging. The domain wall displays the lowest amplitude signals. Since the amplitude signals of different domains have no obvious differences, the domains show a 180° polarization distribution. This also indicates that (HaaOH)BF$_4$ may be a uniaxial molecular ferroelectric. Besides, the clear distribution of as-grown domains has no correlation with

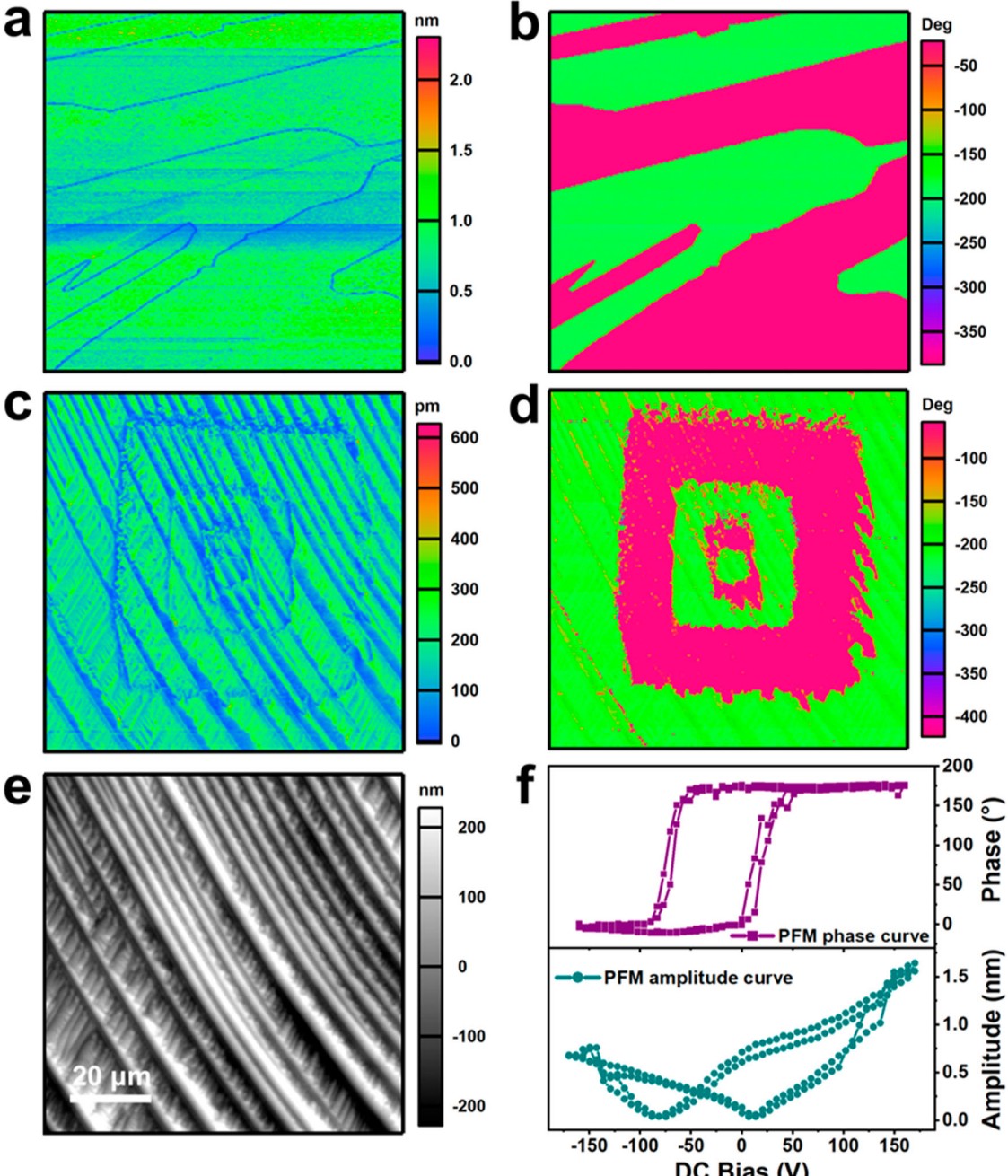

**Fig. 4 | Domain structure and domain switching measurements of the (HaaOH) BF₄ thin film.** The lateral phase (**a**) and lateral amplitude (**b**) of the pristine domain on the as-grown thin film of (HaaOH)BF₄ are displayed. The final state of LPFM amplitude (**c**), phase (**d**), and topography (**e**) images for the 90 × 90 μm² region were observed after $V_{dc}$ was applied in the order of +110 V and −110 V twice to switch fourfold box-in-box domains. **f** Phase and amplitude signals as functions of the tip voltage for a selected point in the off-field period, showing local PFM hysteresis loops.

the surface morphology of the thin film (Supplementary Fig. 12), demonstrating that (HaaOH)BF₄ possesses spontaneous polarization in different directions.

In addition to the as-grown domains and domain walls, ferro-electrics are also characteristic of reversible polarization under an applied electric field. Thanks to the switchable polarization, arbitrary micron patterns of domains can be written on ferroelectrics. Based on this, we characterized the local polarization switching behavior on the (HaaOH)BF₄ thin film through switching spectroscopy PFM (SSPFM) measurement. The reversal of polarization was recorded off-field by PFM phase and amplitude signals when the dc voltage ($V_{dc}$), a trian-gular pulse square wave, and superimposed ac voltage were applied to

the film through the conductive tip (Fig. 4f). Evidently, the PFM phase curve shows a square-shaped hysteresis loop with a 180° contrast under the electric field, and the PFM amplitude curve shows a typical butterfly shape. These curves indicate the switching and hysteresis behaviors of ferroelectric polarization in (HaaOH)BF₄, which are obtained in the off-field period. Meanwhile, temperature-dependent SSPFM measurements indicate that the required voltages for polar-ization switching decrease as the temperature increases and the polarization switching still occurs at 473 K (Supplementary Fig. 13). Furthermore, we selected a 90×90 μm² region and applied the dc bias to the (HaaOH)BF₄ crystal film through the PFM probe to achieve clear domain switching. The selected region on the film is in initial single-

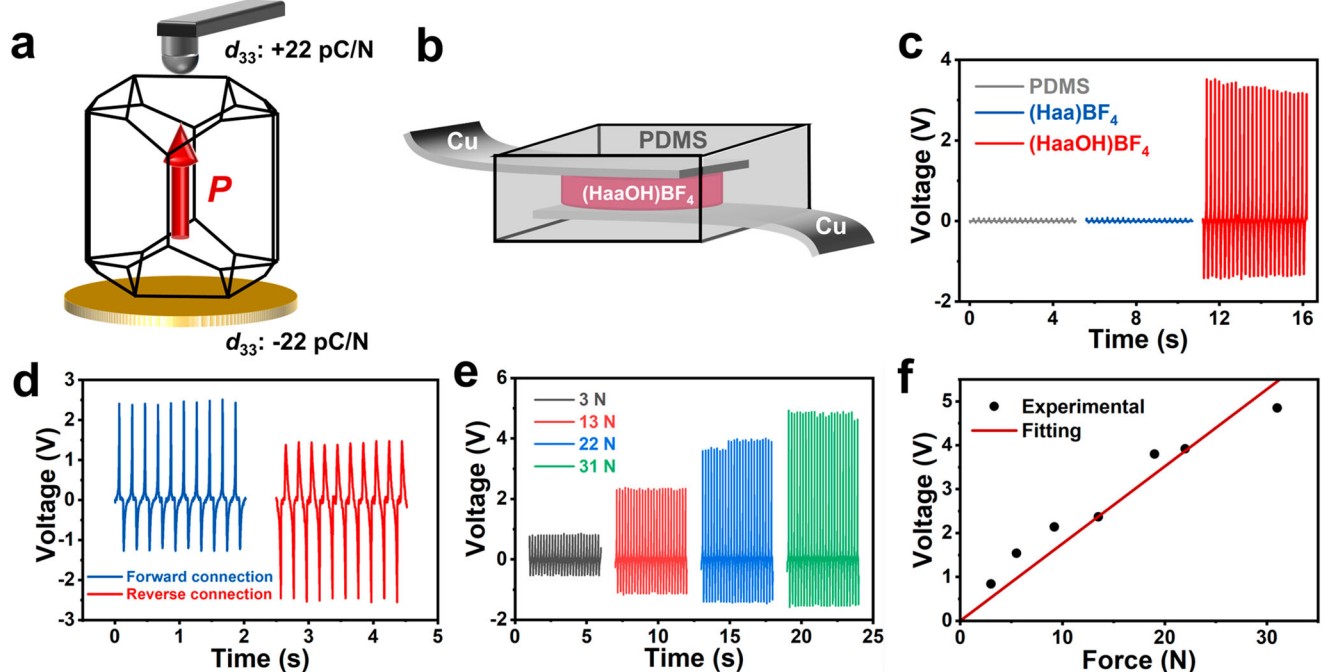

**Fig. 5 | Characterization of piezoelectric properties in (HaaOH)BF₄. a** Diagram of the piezoelectric coefficient $d_{33}$ of the (HaaOH)BF₄ crystal along the [001] direction using the quasi-static method. **b** Schematic illustration of piezoelectric energy-harvesting devices. **c** Generated $V_{OC}$ of energy-harvesting devices based on (HaaOH)BF₄ and (Haa)BF₄ polycrystalline samples under a periodical vertical force of 17 N at a frequency of 10 Hz, and the $V_{OC}$ of the device without samples under the same conditions. **d** $V_{OC}$ of the (HaaOH)BF₄ energy-harvesting device under forward (blue line) and reverse (red line) electrical connections. **e** $V_{OC}$ of the energy-harvesting device containing (HaaOH)BF₄ with different periodical vertical forces applied (3, 13, 22, and 31 N) at a frequency of 10 Hz. **f** Linear fitting of the $V_{OC}$ of the (HaaOH)BF₄ energy-harvesting device as a function of the applied force.

domain state (Supplementary Fig. 14). With the PFM probe, $V_{dc}$ was applied to a selected rectangular region that became smaller and smaller, in the order of +110 V and −110 V twice. The domain in the same region of the film was switched four times under the applied bias at room temperature. Supplementary Fig. 14 vividly shows the meticulous PFM imaging of each polarization reversal process. Evidently, a significant 180° PFM phase contrast exists between different domains, and adjacent domains are separated by clear domain walls with weak amplitude signals. This proves that the polarization in the (HaaOH)BF₄ film can be reversed back and forth, conforming to the principle of ferroelectric polarization. Figure 4c, d displays the fourfold box-in-box domain structures in the final state, which are almost the same as the preset ones. Because the new domain nucleation energy of (HaaOH)BF₄ is lower than the domain growth energy, the domain is easier to nucleate and difficult to diffuse. No obvious change is observed in the morphology of the scanning region before and after domain switching, as shown in Supplementary Fig. 14. And the domain structure written by the electric field can remain stable until 443 K (Supplementary Fig. 15) and for over 2 years at room temperature (Supplementary Figs. 15 and 16). Meanwhile, the (HaaOH)BF₄ film can still be switched under the external electric field at the temperature of 413 K (Supplementary Fig. 17). This rebuts the assumption that domain switching of (HaaOH)BF₄ is caused by charge injection, and further corroborates its intrinsic ferroelectric switching. Furthermore, the written box-in-box domain pattern proves that (HaaOH)BF₄ has switchable spontaneous polarization and can conduct the stable domain configuration on the micron scale.

The resonant PFM mode is widely used for the electro-mechanical coupling of ferroelectric materials, which detects the surface deformation excited by the electric field[17,20]. With PFM, the local piezoresponse of (HaaOH)BF₄ and (Haa)BF₄ thin films can be obtained. The two samples were both driven at the cantilever-sample resonance frequency of 10 V. The amplitude signal obtained is matched with the Damped Simple Harmonic Oscillator model (Supplementary Fig. 18), and the inherent piezoelectric response can be obtained through quality factor correction and resonant amplification. Supplementary Fig. 18 shows the obvious piezoelectric response of (HaaOH)BF₄ and a positive linear relationship between the inherent piezoelectric and the driving voltage. On the contrary, (Haa)BF₄ has no piezoelectric response along different crystal axis directions as expected (Supplementary Fig. 19). The piezoelectric coefficient $d_{33}$ along the corresponding polarization direction of (HaaOH)BF₄ is 22 pC N⁻¹, according to the quasi-static method (Berlincourt method) (Fig. 5a). And the value of $g_{33}$ can be evaluated through the formula of $g_{33} = d_{33}/\varepsilon_{33}$, in which the dielectric permittivity $\varepsilon_{33}$ can be derived from $\varepsilon' = \varepsilon_{33}/\varepsilon_0$ ($\varepsilon' = 21$). Based on the results of $d_{33}$ and $\varepsilon'$ (Supplementary Fig. 11, 19), the $g_{33}$ of (HaaOH)BF₄ is about $165.7 \times 10^{-3}$ Vm N⁻¹, which is higher than that of PZT-based piezoelectric ceramics (about 20 to $40 \times 10^{-3}$ Vm N⁻¹).

To exploit the piezoelectric response with the energy-harvesting capability of (HaaOH)BF₄, we fabricated a device with the structure of electrode-(HaaOH)BF₄-electrode through the package of poly-dimethylsiloxane (PDMS) (Fig. 5b). The device exhibited an open-circuit voltage ($V_{OC}$) of about 3.5 V under a 10 Hz periodic vertical pressure of 17 N. Under the same condition, the blank PDMS device and (Haa)BF₄ device were tested, whose $V_{OC}$ were close to 0 and were lower than (HaaOH)BF₄ device (Fig. 5c). This indicates that the $V_{OC}$ of the (HaaOH)BF₄ device is induced by piezoelectricity. Similarly, switching-polarity tests were conducted to verify that the generated output signals indeed originate from the piezoelectric phenomenon. The reversal polarization test realized by electronic reverse connection also produces corresponding reverse transformation of $V_{OC}$ (Fig. 5d). The reversible electrical signals indicated that the detected outputs were generated by the compression-induced strain of the (HaaOH)BF₄ devices[37]. And this rules out the possibility that $V_{OC}$ comes from the change of system capacitance[38]. Additionally, voltage peak values

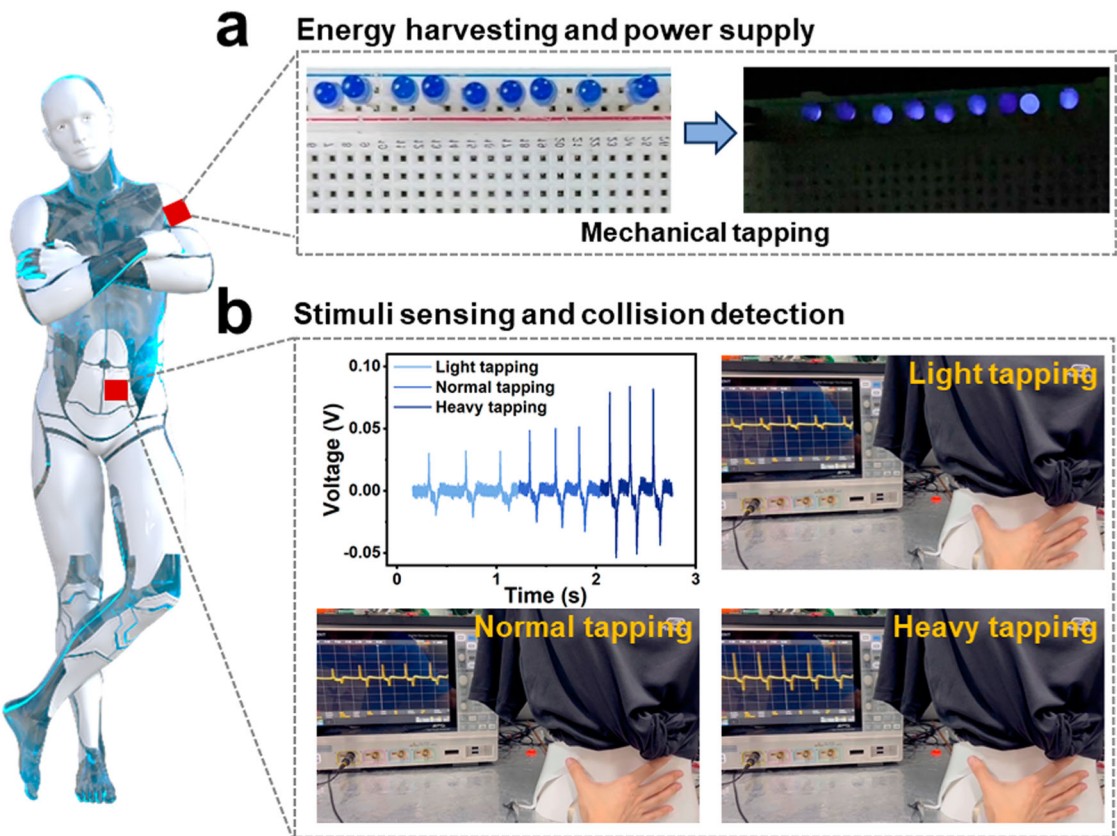

**Fig. 6 | Power supply and stimuli sensing of flexible (HaaOH)BF₄ devices. a** Illuminate LEDs through mechanical tapping driving. **b** Signals of the output voltage in the process of light, normal, and heavy tapping on a dummy. Inset that describes the robot is from *Pixabay* Web site and undergoes a free license.

between compressing and releasing conditions were different and asymmetric. This can be explained by variations in the strain rate during the application and removal of stress on the devices[39]. It is obvious that the $V_{OC}$ of the (HaaOH)BF₄ device increases gradually with the pressure rising from 3 N to 31 N at a stable frequency (Fig. 5e). Figure 5f displays a good linear relationship between the pressure and $V_{OC}$. Meanwhile, the electrical $V_{OC}$ of the device were measured under various force frequencies (Supplementary Fig. 20). The generated electrical output performance is noticeable and capable of responding to different external force frequencies. The output current of the (HaaOH)BF₄ device was measured under a pressure of 20 N, exhibiting a maximum of -0.31 μA (Supplementary Fig. 21). And the piezoelectric device of (HaaOH)BF₄ can realize long-term sensing with the voltage maintained for -5 V after at least 7000 cycles (Supplementary Fig. 22). Furthermore, the output signals of the (HaaOH)BF₄ device at high temperatures have also been verified, which maintained the output voltages of around 2 V at 503 K (Supplementary Fig. 23). This is consistent with the temperature-dependent piezoresponse measurements, which indicate that the sample can maintain piezoelectricity until $T_d$ (Supplementary Fig. 24). This indicates the potential of (HaaOH)BF₄ device for monitoring the working status and vibrations of mechanical components in high-temperature environments. Evidently, the design strategy of hydrogen bond modification can successfully build intermolecular force networks in molecular materials to achieve an orderly molecule arrangement and introduce polarity. And the molecular ferroelectric (HaaOH)BF₄, with a high $T_d$ and stable domains, will not only promote the conversion of mental-free materials into electromechanical converters but also be a promising source for flexible electronic devices.

Due to insufficient development in power and sensing technologies, operating long-duration missions in unstructured environments remain a difficult task for robots[40,41]. Non-metallic (HaaOH)BF₄, possessing good mechanical-to-current conversion capabilities, can be potentially applied to robots to achieve mechanical energy harvesting and self-powered tactile sensing under external mechanical stimuli. Additionally, the output performance of the device was measured by connecting various external load resistors. The output voltage gradually rose as the external load resistance was increased, while the output current decreased (Supplementary Fig. 25a). The obtained maximum output power density was approximately 1.2 μW cm⁻² at a load resistance of $4 \times 10^7$ Ω (Supplementary Fig. 25b). In our proof-of-concept study, the (HaaOH)BF₄ device lit 9 blue LEDs (3.0-3.2 V, 60–64 mW) under the periodic mechanical tapping without using any external power unit (such as a capacitor) (Fig. 6a and Supplementary Movie 1). In addition, by housing the (HaaOH)BF₄ device directly on the robot's surface, it is readily available to detect external mechanical stimuli or collisions. The energy harvester consists of the top conductive adhesive tape, (HaaOH)BF₄, and the bottom conductive adhesive tape, encapsulated by PDMS. The top conductive adhesive tape is electrically grounded (Supplementary Fig. 26). Through this configuration, charges originating from the human body and those induced by the triboelectric effect between the human body and the device can dissipate into the ground. Consequently, the impact of additional charges, apart from the piezoelectric effect, is eliminated. As shown in Fig. 6b, the output piezoelectric response signals vary a lot according to different tapping forces (Supplementary Movie 2), although light tapping can still produce distinct signal responses. Meanwhile, the piezoelectric sensor responded to the pressure quickly, with a fast response time of 10.24 ms under a pressure upon finger tapping (Supplementary Fig. 27). The two applications further prove non-metallic (HaaOH)BF₄-based devices as reliable applicants for robots equipped with energy harvesting and mechanical stimuli sensing.

In summary, we successfully designed the non-ferroelectric (Haa)BF₄ as the molecular ferroelectric (HaaOH)BF₄ through

hydrogen bond modification. The modification of the hydroxyl group has not only introduced ferroelectric polarization but also promoted $T_c$ of (HaaOH)BF$_4$ by at least 336 K. By analyzing the intermolecular force of two compounds' structure, we found that (HaaOH)BF$_4$ obtained more H···O hydrogen bonds and guest-guest interactions. Due to the lattice intermolecular force formed in the crystal, (HaaOH)BF$_4$ has a higher $T_c$, which is even higher than its $T_d$ (528 K). And the ferroelectricity of (HaaOH)BF$_4$ can be maintained until $T_d$, which is close to the high $T_c$ (534 K) as reported among molecular ferroelectrics. This proves hydrogen bond modification as an effective strategy for designing molecular ferroelectrics, stabling ferroelectric domain structure, and optimizing the phase transition temperature. Meanwhile, the fabrication of adamantane with so many modification sites can hopefully optimize the properties of molecular ferroelectrics. Furthermore, stable micron ferroelectric domain structures have been constructed on the film of molecular ferroelectric (HaaOH)BF$_4$. These structures can retain stable until 443 K and for over 2 years at room temperature. The piezoelectric properties of the flexible sample were detected by the energy-harvesting device capable of mechanical energy harvesting and self-powered sensing. The precise molecular design strategy and crystal engineering are crucial to further optimize and promote the development of molecular ferroelectrics. Such an organic molecule with a variety of modification sites also provides opportunities and platforms to enhance modern energies and develop micro-nano electronic devices.

## Methods

### Materials

**Synthesis of single crystals.** All reagents and solvents in the syntheses were of reagent grade and used without further purification. Slight excess of tetrafluoroboric acid (48 wt. % in H$_2$O, 20,12 g, 0.11 mol) and proper amount of deionized water (100 mL) was added to the beaker, and then adamantan-1-amine (15.13 g, 0.1 mol) or 3-amino-1-adamantanol (16.73 g, 0.1 mol) was added to the solution and stirred for 20 min at room temperature. Transparent and colorless crystal of 1-adamantanammonium tetrafluoroborate [(Haa)BF$_4$] or 1-hydroxy-3-adamantanammonium tetrafluoroborate [(HaaOH)BF$_4$] can be obtained by slowly evaporating at room temperature.

**Thin film preparation.** The thin films of [(Haa)BF$_4$] and [(HaaOH)BF$_4$] were prepared by drop-coating approach on ITO/glass substrate. The deionized aqueous solutions of [(Haa)BF$_4$] and [(HaaOH)BF$_4$] with 0.08 g/ml concentration were prepared. And then 15 μL of the solution was dropped on $1 \times 1$ μm$^2$ ITO/glass at 353 K. The transparent films can be obtained after the solution has evaporated.

### Measurements

Methods of single-crystal X-ray crystallography, powder X-ray diffraction, differential scanning calorimetry, thermogravimetric analysis, dielectric measurements, ferroelectric hysteresis loop measurements, piezoresponse force microscopy characterization, piezoelectric coefficient measurements, preparation and measurements of piezoelectric energy-harvesting devices, calculate condition, Hershefield surface analysis and energy framework analysis were described in the Supplementary Information.

### Reporting summary

Further information on research design is available in the Nature Portfolio Reporting Summary linked to this article.

## Data availability

All data generated and analyzed in this study are included in the Article and its Supplementary Information. And all relevant data that support the findings of this study are available from the corresponding authors upon request. The crystal structures generated in this study have been deposited in the Cambridge Crystallographic Data Centre under accession code CCDC: 2192545, 2192429, and 2314039. The data can be obtained free of charge via www.ccdc.cam.ac.uk/data_request/cif, or by emailing data_request@ccdc.cam.ac.uk, or by contacting The Cambridge Crystallographic Data Centre, 12 Union Road, Cambridge CB2 1EZ, UK; fax: +44 1223 336033.

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

## Acknowledgements

This work was supported by the National Key R&D Program of China (Grant No. 2021YFA1200700 (Y.-M.Y.)), the National Natural Science Foundation of China (Grant No. 21925502 (Y.-M.Y.), and 223B2502 (Y.-A.X.)), Postgraduate Research & Practice Innovation Program of Jiangsu Province (No. KYCX23_0224 (Y.-A.X.)), the China Scholarship Council program (Project ID: 202306090116 (Y.-A.X.)) and "the Fundamental Research Funds for the Central Universities, China".

## Author contributions

Y.-M.Y. and J.W. conceived and supervised the project. Y.-A.X. prepared the samples and performed the PFM measurements, piezoelectric energy-harvesting devices, and analysis. S.-S.D. prepared and test the power supply and stimuli sensing of piezoelectric devices. H.-H.H. measured the thermodynamic properties and sorted out the article. J.Y. performed the single-crystal measurement and analysis. Q.P. contributed to *P–V* loop measurements. T.-T.S., X.W., and H.-R.J. participated in the production and performance testing of piezoelectric devices. Y.-M.Y., J.W., Y.-A.X., S.-S.D., and H.-H.H. analyzed the data and results. Y.-A.X., S.-S.D., and H.-H.H. wrote the manuscript with input from all the other authors. All authors discussed the results and contributed to the manuscript preparation.

## Competing interests

The authors declare no competing interests.
