## [Peer Review File · Nature Communications]

Enhancement of Phase Transition Temperature Through Hydrogen Bond Modification in Molecular FerroelectricsREVIEWER COMMENTS

Reviewer #1 (Remarks to the Author):

Molecular ferroelectrics are becoming a hot topic recently. Xiong et al. proposed a hydrogen bond modification method by introducing the hydroxyl group, and boosted the ferroelectric phase transition temperature (T_c). Potential applications of information storage and energy harvesting are demonstrated in the manuscript. I think the work is meaningful, but the overall characterization of electromechanical properties are rough and hence, there are still some problems in the manuscript that need to be explained. Here are my comments.

1. The paper needs to supplement the relevant high temperature experiments to verify the stability of the material at high temperature. For example, dependence of dielectric constant and loss (@1kHz, 10kHz, 100kHz and 1MHz) with temperature range from room temperature to higher than T_c . Relationship of piezoelectric constant and temperature should also be added.
2. The author stated that domain structure can remain stable until 120 °C in Line 269. Why there is no further investigation at higher temperature since T_c is around 255 °C? Can in-situ PFM be finished at various temperature to show domain evolution and its high temperature stability?
3. What is the domain evolution at various temperature
4. Energy harvesting test seems very blurred. No test setup or practical sample are shown in the manuscript, which causes many doubts for the test results.
 - (1) In Fig. 5(d), the force-voltage response waveform is asymmetric, is it attributed to the regular compression force or other reasons? Would you explain the difference between the forward connection and the reverse connection?
 - (2) What is the performance of the energy harvester at various force frequencies, especially at higher frequency?
 - (3) What is the output current? For energy harvesting test, output power and power density are crucial for properties.
 - (4) For the so-called robot energy harvesting test, does the harvester possess the same structure as in Figure 5? Why the output properties seem far lower compared with Figure 5? Also, insulation from tester's hand are important to avoid charge from human body.
 - (5) The application experiments of energy harvesting devices in this paper cannot reflect the high Curie temperature characteristics of this material, and additional experiments are needed to demonstrate the superiority of this material.
5. There are some formatting problems in the article that need to be fixed. The picture annotations in the article should be explained in order of format (a) (b) (c) (d). The legend needs to be given in the Figure 3.
6. Molecular ferroelectrics have been studied for some time and many papers have been published. What is the practical applications for this material compared with other PZT based or lead-free based piezoelectric materials? Especially for the material reported in this manuscript, the piezoelectric properties and T_c can only compared with BaTiO₃, not to mention modified KNN or PZT ceramics.

Reviewer #2 (Remarks to the Author):

The author reported a strategy of hydrogen bond modification to boost the ferroelectric phase transition temperature (T_c). This is important for molecule ferroelectrics in practical applications. I recommend a major revision before the publication. The following are the recommendations.

1. The Sawyer-Tower method should be used to test the hysteresis loop of a single crystal or thin film, and the horizontal coordinate should be the electric field quantity that is more reflective of the coercivity field information of the ferroelectric material, rather than the voltage.
2. The high phase transition temperature is the highlight of this work, however, the piezoelectric application scenarios in this paper are relatively common and can be easily achieved by most ferroelectric materials. The unique high-temperature advantage of the material is not reflected in the application demonstration, and it is essential to use more convincing high-temperature application scenarios to demonstrate the irreplaceability of the material.
3. Although the authors have demonstrated by PFM that (HaaOH)BF₄ remains ferroelectric at 393 K, there is still a long way to go before its phase transition temperature (>528 K). Please add the variable-temperature electric hysteresis loop and variable-temperature d₃₃ before the thermal decomposition temperature to increase the convincing data at the experimental level.
4. The authors tested that (Haa)BF₄ does not have a piezoelectric effect (d₃₃=0), yet there is still a clear electrical signal response in Fig. 5(c), which is contradictory, please check the experimental data.
5. More detailed experimental details and test instrument models need to be given: including hysteresis loop, dielectric constant, piezoelectric coefficient tests, and energy harvesting applications.
6. Does d₃₃ vary along different crystal axis directions? The authors are suggested to give the relevant experimental data.
7. The coordinate numbers in some of the figures are so small that they are not readable, so please standardize the format.

Reviewer #3 (Remarks to the Author):

This manuscript mentions the application of hydrogen bond modification design strategy to introduce spontaneous polarization and increase the phase transition temperature in molecular materials. The introduction of hydroxyl group on (Haa)BF₄ enabled the formation of intermolecular hydrogen bond network and achieved the synthesis of a new molecular ferroelectric (HaaOH)BF₄. Meanwhile, the intermolecular hydrogen bond can stable the polarization in molecular ferroelectric for a longer period of time. Based on (HaaOH)BF₄, the manuscript also explores the production of piezoelectric energy-harvesting devices and the application in self-powered sensing.

The hydrogen bond modification mentioned in the manuscript is a very effective strategy for introducing polarization and optimizing performance in molecular materials. And the new molecular ferroelectric (HaaOH)BF₄ also has good properties and device performance. The crystal engineering strategy and the molecular ferroelectric provide new opportunities and platforms for the development of modern energy and micro-nano electronic devices. Therefore, I think this manuscript is suitable for publication in Nature Communications with few revisions and improvements.

1. (Haa)BF₄ undergoes a structural phase transition at 192 K, and the crystal structure will be different from room temperature. Thus, it is necessary to supplement the crystal structure of (Haa)BF₄ at low temperature and add some analysis and discussion.
2. The manuscript mentioned "the highest enhancement of phase transition temperature". So, can the author add a statistical table showing the enhancement of phase transition

temperature in molecular ferroelectrics? And the table may include compounds, different methods and the value of enhancement.

3. There are obvious differences between the crystals of (HaaOH)BF₄ and (Haa)BF₄ in Figure S1. The crystal growth habits of (Haa)BF₄ can also be simulated by BFDH method and compared with (HaaOH)BF₄ and the actual crystal morphology.

4. Why the powder tablets of (Haa)BF₄ are a little transparent, but the powder tablets of (HaaOH)BF₄ is not?

5. Please add the process of making piezoelectric energy-harvesting devices and the sensing devices in Methods of the manuscript.

6. Response time is a very important parameter in piezoelectric sensors. So, what is the value of the response time of the piezoelectric sensors based on the molecular ferroelectric?

Author's response

Authors' response to all reviewers:

First of all, we would like to express our greatest appreciation to all reviewers for their affirmation of the academic innovations in our manuscript. Moreover, we are grateful to reviewers for their questions and suggestions, which help us to improve the quality of this manuscript as well as our future research indeed. After carefully studying the comments from all reviewers, we have refined the manuscript by completing new experiments, adding new analytical discussions and supporting data analysis. The addition and revision of the manuscript include: new temperature-dependent PFM measurements, device performance measurements, piezoelectric measurements, crystal structural measurements and dielectric measurements, etc. We have also added more analysis, discussion and methods to support our claims and improve the scientific importance. We also apologize for typos, minor errors, confusion and ambiguities in the previous version of the manuscript. In this revision, we have thoroughly proofread the manuscript and improved the quality of the manuscript and figures in terms of scientific writing and legends.

Below is the point-to-point response to each reviewer's comments and concerns. (Authors' response was colored in blue and quote of addition/revision of manuscript was colored in purple.)

Reviewer #1 (Remarks to the Author):

Molecular ferroelectrics are becoming a hot topic recently. Xiong et al. proposed a hydrogen bond modification method by introducing the hydroxyl group, and boosted the ferroelectric phase transition temperature (T_c). Potential applications of information storage and energy harvesting are demonstrated in the manuscript. I think the work is meaningful, but the overall characterization of electromechanical properties are rough and hence, there are still some problems in the manuscript that need to be explained. Here are my comments.

1. The paper needs to supplement the relevant high temperature experiments to verify

the stability of the material at high temperature. For example, dependence of dielectric constant and loss (@1kHz, 10kHz, 100kHz and 1MHz) with temperature range from room temperature to higher than T_c . Relationship of piezoelectric constant and temperature should also be added.

Authors' response: The comments are meaningful and helpful for us to refine and improve the content of the manuscript. We have incorporated additional high-temperature experiments to enhance the stability assessment of the material. Specifically, we conducted temperature-dependent measurements of the dielectric constant and loss at frequencies of 500 Hz, 1kHz, 5kHz, 10kHz, 100kHz, and 1MHz, as well as the temperature-dependent piezoelectric constant. Through differential scanning calorimetry (DSC) measurement and thermogravimetric analysis, we determined the decomposition temperature of (HaaOH)BF₄ to be 528 K. Notably, no structural phase transition was observed below this temperature. Therefore, we heat the sample as close to the temperature of 528 K when measuring the temperature-dependent dielectric constant, loss and piezoelectric constant. The dielectric constant and loss in the crystal sample of (HaaOH)BF₄ were probed by the temperature- and frequency-dependent dielectric permittivity measurements across various frequencies ranging from 500 Hz to 1 MHz (Figure R1). The real part (ϵ') and the imaginary part (ϵ'') of the dielectric constant were found to gradually increase upon increasing the temperature. The observed trends in the ϵ' values indicate the absence of phase transition in (HaaOH)BF₄ as noted earlier from its DSC profile. The absence of structural phase transitions in (HaaOH)BF₄ can be attributed to the presence of huge (HaaOH)⁺ cations and their effective intricate hydrogen bond interactions with BF₄⁻ anions. The additional experiment results have been added to the Supplementary Information (Figure S10), and the analysis and description have been supplemented in the revised manuscript as: "Meanwhile, the dielectric constant and loss in (HaaOH)BF₄ were probed by the temperature- and frequency-dependent dielectric permittivity measurements across various frequencies ranging from 500 Hz to 1 MHz (Figure S10). The real part (ϵ') (Figure S10a) and the imaginary part (ϵ'') (Figure S10b) of the dielectric constant were

found to gradually increase upon increasing the temperature and absent of structural phase transition.”

Figure S10 (R1). Variable-temperature and variable-frequency complex dielectric constant of (HaaOH)BF₄ measured on the single-crystal along *c*-axis. Temperature-dependent dielectric constant of real part ϵ' (a) and imaginary part ϵ'' (b). Frequency-dependent dielectric at 293 K (c) and at different temperatures (d). The complex dielectric constant $\epsilon = \epsilon' - i\epsilon''$, in which ϵ' and ϵ'' are the real and imaginary parts, respectively. The insets in (a) and (b) show the temperature-dependent ϵ' and ϵ'' at the frequency of 1000 kHz, respectively.

The temperature-dependent piezoelectric coefficient (d_{33}) was measured on the as-grown single-crystal of (HaaOH)BF₄ through the quasi-static method. The d_{33} coefficient remained stable at around 22 pC/N along the proximity of the [001] direction of the crystal until approximately 515 K (Figure R2). This indicates that (HaaOH)BF₄ can maintain the noncentrosymmetric structure and piezoelectricity until the decomposition temperature. Meanwhile, we used the piezoresponse force microscopy (PFM) technique to measure the temperature-dependent local piezoresponse of

(HaaOH)BF₄ films. The acquired curves exhibited typical resonance peaks which fitted well by the simple harmonic oscillator model. The effective amplitude was determined through dividing the amplitude by the quality factor. It was observed that (HaaOH)BF₄ films still exhibited significant piezoresponse up to 513 K (Figure R2). This is consistent with the result of the temperature-dependent d_{33} measurement. The additional experiment results have been added to the Supplementary Information (Figure S23), and the analysis and description have been supplemented in the revised manuscript as: “This is consistent with the temperature-dependent piezoresponse measurements, which indicate that the sample can maintain piezoelectricity until T_d (Figure S23).”

Figure S23 (R2). Temperature-dependent piezoelectric properties of (HaaOH)BF₄. (a) Piezoelectric coefficient (d_{33}) of (HaaOH)BF₄ at different temperatures. (b) PFM resonance peaks at different temperatures.

2. The author stated that domain structure can remain stable until 120 °C in Line 269. Why there is no further investigation at higher temperature since T_c is around 255 °C? Can in-situ PFM be finished at various temperature to show domain evolution and its high temperature stability?

Authors' response: The high-temperature in-situ PFM measurements are significant influenced by thermal disturbance, affecting the accuracy and stability of PFM imaging and signals. Thermal measurements confirmed that (HaaOH)BF₄ did not undergo structural phase transition before the decomposition temperature and should remain in the ferroelectric phase. Based on this, we have tried to increase the temperature of in-

situ PFM measurement to show domain evolution and its high temperature stability. We performed PFM imaging at increasing temperature intervals of 50 K in a $90 \times 90 \mu\text{m}^2$ region that has already been switched. As shown in Figure R3, the domain structure can remain stable up to 443 K. The PFM imaging domain structure at 443 K can prove the stability of ferroelectricity and domain structures in $(\text{HaaOH})\text{BF}_4$. However, the quality, accuracy and testing process of PFM imaging were severely disrupted as the temperature increases, making it impossible to obtain images at higher temperatures. The additional experiment results have been added to the Supplementary Information (Figure S14), and the analysis and description have been supplemented in the revised manuscript as: “And the domain structure written by the electric field can remain stable until 443 K (Figure S14) and for over 2 years at room temperature (Figure S14-S15).”

Figure S14 (R3). Evolution of the domain structure of (HaaOH)BF₄ at various temperatures. Lateral PFM phase and amplitude images (a, b) at 293 K, (c, d) at 343 K, (e, f) at 393 K and (g, h) at 443 K, respectively.

3. What is the domain revolution at various temperature

Authors' response: We conducted domain revolution measurements at various temperatures to demonstrate domain switching and the retention of ferroelectricity at high temperatures. We selected a $30 \times 30 \mu\text{m}^2$ single domain region and heated the (HaaOH)BF₄ thin film to 373 K (Figure R4a, b). By applying a DC voltage of -160 V through the probe, a small portion of the domain within the black solid rectangular box was switched (Figure R4c, d). As the temperature increased to 393 K, the area of the switched domain expanded under the same voltage applied through the probe in the same condition (Figure R4e, f). As the temperature continued to rise to 413 K, larger area obvious domains were switched after applying -160 V DC voltage in the same manner (Figure R4g, h). These switched domains tended to grow along the polarization direction, which is related to the energy of different types of domain walls. The temperature-dependent polarization switching process indicates that the coercive field of the material decreases with increasing temperature, facilitating easier domain switching and larger area domain growth. Additionally, this also demonstrates that the spontaneous polarization of (HaaOH)BF₄ thin films can still be switched under the external electric field at 413 K. The additional experiment results have been added to the Supplementary Information (Figure S16), and the analysis and description have been supplemented in the revised manuscript as: "Meanwhile, the (HaaOH)BF₄ film can still be switched under the external electric field at the temperature of 413 K (Figure S16)."

Figure S16 (R4). Revolution of the domain at various temperature. The original PFM phase (a) and amplitude (b) at the selected $30 \times 30 \mu\text{m}^2$. The domain revolution under the same electric polarization of -160 V DC voltage at (c, d) 373 K, (e, f) 393 K and (g, h) 413 K.

4. Energy harvesting test seems very blurred. No test setup or practical sample are shown in the manuscript, which causes many doubts for the test results.

Authors' response: We appreciate the reviewer for the suggestions and the comments are significant in energy harvesting devices measurement. We conducted additional experiments and provided supplementary descriptions for the test setup, practical samples and device performance in response to reviewer's comments. The device architecture, schematic diagrams of the test setup and additional experiments were added to the Supplementary Information. Additionally, the experimental processes and setup have been supplemented in the Methods section. The detailed responses and modifications were provided for each question below.

(1) In Fig. 5(d), the force-voltage response waveform is asymmetric, is it attributed to the regular compression force or other reasons? Would you explain the difference between the forward connection and the reverse connection?

Authors' response: The comment is significant in piezoelectric devices of the force-voltage response waveform. The positive and negative voltage peak values were different and asymmetric between compressing and releasing conditions. This can be explained by the difference of the strain rate in the process of applying and removing

the pressure on the devices. (DOI: 10.1002/adfm.202011073) Meanwhile, the devices compressed quickly under the pressure but rebounded slowly when the pressure was withdrawn, which lead the force-voltage response waveform to be asymmetric.

Switching-polarity tests can be used to demonstrate the generated output signals originated from the piezoelectric phenomenon. The positive output signals were observed when the device was connected in forward connection (Figure R5). While under a reverse connection, opposite output signals were measured as shown in Figure R5. It is obvious that the electric signals are reversible, indicating that the detected outputs are produced from the (HaaOH)BF₄ devices strained by compressing motion. (10.1016/j.nanoen.2017.05.010)

The analysis and description have been supplemented in the revised manuscript as: “Similarly, switching-polarity tests were conducted to verify that the generated output signals indeed originate from the piezoelectric phenomenon. The reversal polarization test realized by electronic reverse connection also produces corresponding reverse transformation of V_{OC} (Figure 5d). The reversible electrical signals indicated that the detected outputs were generated by the compression-induced strain of the (HaaOH)BF₄ devices. (DOI: 10.1016/j.nanoen.2017.05.010) And this rules out the possibility that V_{OC} comes from the change of system capacitance. (DOI: 10.1063/1.3072362) Additionally, voltage peak values between compressing and releasing conditions were different and asymmetric. This can be explained by variations in the strain rate during the application and removal of stress on the devices. (DOI: 10.1002/adfm.202011073)”

Figure 5d (R5). V_{OC} of the (HaaOH)BF₄ energy-harvesting device under forward (blue line) and reverse (red line) electrical connections.

(2) What is the performance of the energy harvester at various force frequencies, especially at higher frequency?

Authors' response: The comment on the performance of energy harvesting devices at various force frequencies is crucial for characterizing of the devices. We conducted tests on the output signals of the (HaaOH)BF₄ device at both low and high force frequencies. The electrical V_{OC} of the (HaaOH)BF₄ device were measured at low applied frequencies of 1 Hz, 3 Hz and 5 Hz under a constant applied force of 20 N. The generated output V_{OC} were almost the same under different frequencies (Figure R6). This suggests that the electrical output performance of energy harvester is stable at low frequency. Meanwhile, the electrical V_{OC} of the (HaaOH)BF₄ device were measured at high applied frequencies of 40 Hz, 50 Hz and 60 Hz (Figure R6). However, the custom-design mechanical system was unable to apply the preset force to the device effectively as the frequency increasing. And the force sensor was also unable to effectively collect the magnitude of the pressure. As a result, the decrease in the effectively applied force led to a reduction in the output signals of the device with the increase in high frequency.

Nevertheless, the device still maintained a noticeable signal output capability. This demonstrates that the energy harvester also exhibits a noticeable output response at high frequencies. The additional experiment results have been added to the Supplementary Information (Figure S19), and the analysis and description have been supplemented in the revised manuscript as: “Meanwhile, the electrical V_{OC} of the device were measured under various force frequencies (Figure S19). The generated electrical output performance is noticeable and capable of responding to different external force frequencies.”

Figure S19 (R6). The performance of the (HaaOH)BF₄ energy harvester at various force frequencies. (a) The response signals at low frequency of 1 Hz, 3 Hz and 5 Hz. (b) The response signals at high frequency of 40 Hz, 50 Hz and 60 Hz.

(3) What is the output current? For energy harvesting test, output power and power density are crucial for properties.

Authors' response: The comments on the performance of energy harvesting devices at output current, output power and power density are crucial for characterizing of the devices. Firstly, we measured the output current for energy harvesting and the (HaaOH)BF₄ device exhibited a maximum output current of $\sim 0.31 \mu\text{A}$ under a pressure of 20 N, as depicted in Figure R7. Furthermore, to identify the effect of proper impedance matching of external load resistance on the output electrical performance of the (HaaOH)BF₄ device, its output electrical responses such as open-circuit voltage

(V_{OC}), short-circuit current (I_{SC}), and power density of device were measured at different load resistance (R_L) values from 1×10^5 to $3 \times 10^9 \Omega$ (Figure R8). The electrical output voltage and current values were measured at different R_L . The output V_{OC} was gradually enhanced by increasing the R_L from 10^5 to $10^9 \Omega$ and reached the highest value of ~ 13 V (Figure R8a). Meanwhile, the I_{SC} followed the reverse trend, exhibiting a maximum electrical output value of $\sim 0.29 \mu\text{A}$ at low R_L , and gradually decreased and close to $0 \mu\text{A}$ at high R_L (Figure R8a). The effective power density (W_D) of the device was determined by $W_D = U^2 R_L^{-1} A^{-1}$, where U represents the output value of the device and A represents the active area of the device (0.92 cm^2). The power density varies with changes in the load resistance and the highest electrical output power density of $\sim 1.2 \mu\text{W cm}^{-2}$ was achieved by the device at the R_L of $4 \times 10^7 \Omega$ (Figure R8b). The additional experiment results have been added to the Supplementary Information (Figure S20, S24), and the analysis and description have been supplemented in the revised manuscript as: “The output current of the (HaaOH)BF₄ device was measured under a pressure of 20 N, exhibiting a maximum of $\sim 0.31 \mu\text{A}$ (Figure S20).”

“And the output performance of the device was measured by connecting various external load resistors. The output voltage gradually rose as the external load resistance was increased, while the output current decreased (Figure S24a). The obtained maximum output power density was approximately $1.2 \mu\text{W cm}^{-2}$ at a load resistance of $4 \times 10^7 \Omega$ (Figure S24b).”

Figure S20 (R7). The generated output current values of the (HaaOH)BF₄ device under 20 N pushing force.

Figure S24 (R8). (a) The load resistance dependent output voltage and current values and (b) output power density measured at various external load resistances values ranging from 1×10^5 to $3 \times 10^9 \Omega$.

(4) For the so-called robot energy harvesting test, does the harvester possess the same structure as in Figure 5? Why the output properties seem far lower compared with

Figure 5? Also, insulation from tester's hand are important to avoid charge from human body.

Authors' response: We appreciate the reviewer for the suggestions and the comments are significant in energy harvesting devices measurement. The structure of the harvester is the same as the device shown in Figure R10. The $(\text{HaaOH})\text{BF}_4$ is sandwiched between two conductive adhesive tapes to construct the energy harvester, where the top conductive adhesive tape is electrically connected to the ground during the process of tapping. And we have added the Figure R9 to illustrate the device and testing circuit structure.

Figure R9. Device structure of $(\text{HaaOH})\text{BF}_4$ and sensor testing circuit.

The output signals of piezoelectric sensor in Figure R11 were about 0.08 V under tapping with a finger. And this output properties seem far lower than the output voltage in Figure R10 under mechanical pressure. We attribute this phenomenon to two factors. Firstly, the force applied by finger tapping (< 1 N) is significantly lower than that applied by mechanical equipment. However, the output voltage in Figure R10c was about 3 V which was measured under a periodical vertical force of 17 N. As shown in Figure R10f, the output voltage will be less than 0.17 V when the pressure applied to the device is less than 1 N. Secondly, there was a loss of charges induced by mechanical force on the top electrode layer due to its electrical connection to the ground during the sensing measurement. Therefore, the output signals of the device were relatively low under tapping conditions. And this also reflects good sensitivity of the device even under weak force tapping.

Figure 5 (R10). Characterization of piezoelectric properties in (HaaOH)BF₄. (a) Diagram of the piezoelectric coefficient d_{33} of the (HaaOH)BF₄ crystal along the [001] direction using the quasi-static method. (b) Schematic illustration of piezoelectric energy-harvesting devices. (c) Generated V_{OC} of energy-harvesting devices based on (HaaOH)BF₄ and (Haa)BF₄ polycrystalline samples under a periodical vertical force of 17 N at a frequency of 10 Hz, and the V_{OC} of the device without samples under the same conditions. (d) V_{OC} of the (HaaOH)BF₄ energy-harvesting device under forward (blue line) and reverse (red line) electrical connections. (e) V_{OC} of the energy-harvesting device containing (HaaOH)BF₄ with different periodical vertical forces applied (3, 13, 22, and 31 N) at a frequency of 10 Hz. (f) Linear fitting of the V_{OC} of the (HaaOH)BF₄ energy-harvesting device as a function of the applied force.

Figure 6 (R11). Power supply and stimuli sensing of flexible (HaaOH)BF₄ devices. (a) Illuminate LEDs through mechanical tapping driving. (b) Signals of the output voltage in the process of light, normal and heavy tapping on a dummy.

It is crucial to insulate the tester's hand to prevent interference from the human body's charge. In our device setup, the encapsulation with the polymer polydimethylsiloxane (PDMS) effectively prevents direct contact between the human body and the electrodes, thereby preventing charge transfer. Meanwhile, in our testing setup, we designed corresponding component to eliminate the potential impact through the electrical connection to the ground of the top conductive adhesive tapes.

More detailed description about harvester design and discussion has been added in the revised manuscript as: “The energy harvester consists of the top conductive adhesive tape, (HaaOH)BF₄, and the bottom conductive adhesive tape, encapsulated by PDMS. The top conductive adhesive tape is electrically grounded (Figure S25). Through this configuration, charges originating from the human body and those induced by the triboelectric effect between the human body and the device can dissipate into the ground.

Consequently, the impact of additional charges, apart from the piezoelectric effect, is eliminated.”

(5) The application experiments of energy harvesting devices in this paper cannot reflect the high Curie temperature characteristics of this material, and additional experiments are needed to demonstrate the superiority of this material.

Authors’ response: The comment on the performance of energy harvesting devices at high temperatures is crucial for the manuscript. In some extreme conditions, high-temperature-resistant self-powered sensors can be used for monitoring the working status of mechanical components. For example, in high temperature operating environment at around 200 °C such as deep earth engineering, petroleum refining pipelines and near-earth space stations, self-powered sensors can be used for vibration monitoring of mechanical components like pipelines and monitoring the operational status of machinery components such as engines. Therefore, it is crucial for verifying the operation of self-powered devices at high temperature to adapt to high-temperature environment applications.

Due to the inability to combine the mechanical press platform with the heating system, we have redesigned the experimental setup. The device was placed on a heating apparatus, and the actual temperature of the sample was measured using a thermometer. Simultaneously, the output signals under different temperatures were collected by manually controlling a mechanical device tapping. As shown in Figure R12, the increase in temperature leads to a slight decline in the device's performance compared to output signals at 293 K. This could be induced by the less secure connection between the conductive adhesive tape and the sample at high temperatures as well as softening of the polymer. However, the device maintained good output performance as the temperature continues to rise to 503K (Figure R12). This demonstrates the stability of (HaaOH)BF₄ and the performance of the device at high temperatures. The additional experiment results have been added to the Supplementary Information (Figure S22), and the analysis and description have been supplemented in the revised manuscript as: “Furthermore, the output signals of the (HaaOH)BF₄ device at high temperatures have

also been verified, which maintained the output voltages of around 2 V at 503 K (Figure S22). This indicates the potential of (HaaOH)BF₄ device for monitoring the working status and vibrations of mechanical components in high-temperature environments.”

Figure S22 (R12). The output signals of the (HaaOH)BF₄ piezoelectric device at different temperatures.

5. There are some formatting problems in the article that need to be fixed. The picture annotations in the article should be explained in order of format (a) (b) (c) (d). The legend needs to be given in the Figure 3.

Authors' response: The comments are meaningful and helpful for us to improve the manuscript in terms of scientific writing. We have revised the picture annotations in the manuscript and adjusted the order of labeling in Figure 1 and Figure 2 to explain them in the order of format (a) (b) (c) (d). We have modified the color of the vertical axes in Figure 3 (R13) to correspond with the data. And the guidance has been provided

through arrows. All modified images, annotations and legends have been revised in the manuscript.

Figure 3 (R13). J - V curve and P - V hysteresis loop of the (HaaOH)BF₄ film at room temperature.

6. Molecular ferroelectrics have been studied for some time and many papers have been published. What are the practical applications for this material compared with other PZT based or lead-free based piezoelectric materials? Especially for the material reported in this manuscript, the piezoelectric properties and T_c can only be compared with BaTiO₃, not to mention modified KNN or PZT ceramics.

Authors' response: The alterations in spontaneous polarization direction and intensity under external fields (such as stress or electric field), coupled with concurrent variations in unit cell parameters, are pivotal mechanisms accounting for the high piezoelectric response in ferroelectric materials. (DOI: 10.7498/aps.69.20200980) However, achieving a high piezoelectric response necessitates a simultaneous consideration of both the dielectric constant and the strength of spontaneous polarization. The existing research results indicate that the T_c of lead-based ceramics with the maximum piezoelectric effect is around 373 K. (DOI: 10.1007/s10832-007-9047-0) And the T_c of potassium sodium niobate (KNN) based ceramics with the maximum piezoelectric

effect is around 423 K. (DOI: 10.1021/cr5006809) Compared to this, the material reported in this manuscript maintains ferroelectricity at higher temperature (528 K), and the energy harvesting devices also exhibit distinct output signals at higher temperatures. Meanwhile, the devices exhibit stable output signals under both low-frequency and high-frequency force driving. This is applicable to some special application scenarios such as vibration monitoring and operational status monitoring of mechanical components in deep geological exploration, near-Earth space stations, petroleum refining pipelines, and engine instrumentation. Meanwhile, as shown in the manuscript, we have also explored the device in self-powered sensing applications.

At present, molecular ferroelectrics cannot replace inorganic materials in most devices and applications. However, under some specific requirements, such as flexibility, low acoustic impedance matching, biocompatibility, etc., molecular ferroelectric materials can serve as complements to inorganic materials. Due to the low density and the inclusion of organic components, molecular ferroelectrics typically exhibit a lower acoustic impedance than inorganic ferroelectrics. Especially in the case of molecular ferroelectrics with non-metallic components, the acoustic impedance typically ranges around $3 \times 10^6 \text{ kg m}^{-2} \text{ s}^{-1}$. This makes molecular ferroelectrics more compatible with water and biological tissues, rendering them more suitable for applications in underwater acoustics and medical fields. Furthermore, the value of piezoelectric voltage coefficient (g) directly represents material figure of merit for piezoelectric sensors. Although molecular ferroelectrics typically have lower dielectric constants than inorganic materials, this would result in molecular ferroelectrics having superior piezoelectric voltage coefficients compared to inorganic ceramics (such as the piezoelectric voltage coefficient g_{33} of PZT ceramics is usually in the range from 20 to $40 \times 10^{-3} \text{ V m N}^{-1}$). This also makes molecular ferroelectrics promising candidates for next-generation smart piezoelectric sensors.

More importantly, molecular ferroelectrics can achieve optimization and breakthroughs of performance through the design of cations and anions, due to the flexibility of structural design. For example, single-component piezoelectric material TMCM-CdCl₃ (TMCM, trimethylchloromethyl ammonium) with high d_{33} of 383 pC N⁻¹ and two-

component piezoelectric material $(\text{TMFM})_x(\text{TMCM})_{1-x}\text{CdCl}_3$ near morphotropic phase boundaries ($x = 0.26$) with d_{33} of 1540 pC N^{-1} have been synthesized in molecular ferroelectric materials. (DOI: 10.1126/science.aai8535; 10.1126/science.aav3057; 10.1021/jacs.2c11213) Similarly, the design strategy of constructing a hydrogen bonding network through hydroxyl group modification in organic molecular ferroelectrics can effectively introduce ferroelectric polarization and enhance the phase transition temperature in this manuscript. The hydrogen bond modification strategy will also guide the design of more molecular ferroelectric materials with diverse structures, high phase transition temperatures, polarization stability, and high piezoelectric coefficients. Therefore, the performance of molecular ferroelectrics will be optimized and enhanced through in-depth research on chemical design strategies, structural regulation, and functional applications. Molecular ferroelectrics will be further understood and ultimately achieve practical applications.

Based on this, we have refined the description in the revised manuscript as: “The current research on molecular ferroelectrics is focused on new materials, methodological advancements, and improving performance, etc. The practical applications of molecular ferroelectrics still need further exploration. Meanwhile, molecular ferroelectrics can serve as a complement to inorganic and polymeric materials in certain specialized applications.”

Reviewer #2 (Remarks to the Author):

The author reported a strategy of hydrogen bond modification to boost the ferroelectric phase transition temperature (T_c). This is important for molecule ferroelectrics in practical applications. I recommend a major revision before the publication. The following are the recommendations.

1. The Sawyer-Tower method should be used to test the hysteresis loop of a single crystal or thin film, and the horizontal coordinate should be the electric field quantity that is more reflective of the coercivity field information of the ferroelectric material, rather than the voltage.

Authors' response: The comment is significant for ferroelectric hysteresis loop in testing methods and ferroelectric properties. Traditional ferroelectric P - E hysteresis loop measurement is mainly based on single crystal samples, but high-quality molecular ferroelectric single crystal growth and directional cutting require a long time. Therefore, we pay more attention to the properties of thin films than single crystals. Molecular ferroelectric films offer advantages such as easy preparation, low cost and mechanical flexibility, making them ideal for property characterization and application. Thus, we conducted measurements on molecular ferroelectric thin films to characterize their properties. Currently, two methods have been developed and introduced to measure hysteresis loops, i.e., the Sawyer-Tower method and the double-wave method. The Sawyer-Tower method, also known as the compensation method, is based on the capacitor model circuit. And the hysteresis loop is obtained directly by means of parameter modulation (DOI: 10.1111/jace.12773). However, the parallel plate capacitor model of the Sawyer-Tower method is not well-suited for testing ferroelectric films, especially films with in-plane polarization distribution. This necessitates larger electrodes and specialized construction methods to establish a model of a flat capacitor. Due to the in-plane distribution of polarization in $(\text{HaaOH})\text{BF}_4$ thin films, larger electrodes and specialized construction methods are necessary to establish a parallel plate capacitor for testing purposes. However, this approach may render the material

more susceptible to breakdown and necessitate the application of higher voltages. Therefore, we employed the double-wave method to measure the hysteresis loop of the (HaaOH)BF₄ film. The double-wave method is based on a resistance model circuit, which distinguishes undesirable components from the ferroelectric component by applying a unipolar waveform twice, enabling the undesirable components to be subtracted automatically (DOI: 10.1143/JPSJ.77.064706). In the first triangular wave, three types of charges Q_f , Q_d and Q_c generated by ferroelectricity, dielectric and conductive components, respectively, all contribute to the total charge Q . In the second triangular wave, only Q_d and Q_c components exist because the polarization has already switched. Consequently, the Q_f part which corresponds to the contribution of ferroelectric properties can be separated through mathematical manipulation. Therefore, we measured the in-plane hysteresis loop of the (HaaOH)BF₄ thin film using the double-wave method (Figure 3). Furthermore, we validated the ferroelectricity of (HaaOH)BF₄ through piezoresponse force microscopy (PFM) and switching spectroscopy PFM (SSPFM) for more intuitive verification (Figure 4).

Because the double-wave method relies on the resistance model, determining the electric field distribution during testing of films with in-plane polarization is challenging. Therefore, the horizontal coordinate of the hysteresis loop measured by double-wave method is generally represented as the voltage quantity (DOI: 10.1126/science.aai8535; 10.1038/s41467-022-33925-2). Although we can estimate the electric field roughly by measuring the distance between the electrodes and represent the horizontal coordinate with the electric field quantity (Figure R14), accurately obtaining the actual value of the electric field is difficult. Furthermore, we are continuing efforts to grow large single crystals of (HaaOH)BF₄, although this process will require additional time.

Figure R14. J - E curve and P - E hysteresis loop of the $(\text{HaaOH})\text{BF}_4$ film at room temperature.

2. The high phase transition temperature is the highlight of this work, however, the piezoelectric application scenarios in this paper are relatively common and can be easily achieved by most ferroelectric materials. The unique high-temperature advantage of the material is not reflected in the application demonstration, and it is essential to use more convincing high-temperature application scenarios to demonstrate the irreplaceability of the material.

Authors' response: The comment on the performance of energy harvesting devices at high temperatures is crucial for the manuscript. In some extreme conditions, high-temperature-resistant self-powered sensors can be used for monitoring the working status of mechanical components. For example, in high temperature operating environment at around 200 °C such as deep earth engineering, petroleum refining pipelines and near-earth space stations, self-powered sensors can be used for vibration monitoring of mechanical components like pipelines and monitoring the operational status of machinery components such as engines. Therefore, it is crucial for verifying

the operation of self-powered devices at high temperature to adapt to high-temperature environment applications.

Due to the inability to combine the mechanical press platform with the heating system, we have redesigned the experimental setup. The device was placed on a heating apparatus, and the actual temperature of the sample was measured using a thermometer. Simultaneously, the output signals under different temperatures were collected by manually controlling a mechanical device tapping. As shown in Figure R15, the increase in temperature leads to a slight decline in the device's performance compared to output signals at 293 K. This could be induced by the less secure connection between the conductive adhesive tape and the sample at high temperatures as well as softening of the polymer. However, the device maintained good output performance as the temperature continues to rise to 503K (Figure R15). This demonstrates the stability of (HaaOH)BF₄ and the performance of the device at high temperatures. The additional experiment results have been added to the Supplementary Information (Figure S22), and the analysis and description have been supplemented in the revised manuscript as: “Furthermore, the output signals of the (HaaOH)BF₄ device at high temperatures have also been verified, which maintained the output voltages of around 2 V at 503 K (Figure S22). This indicates the potential of (HaaOH)BF₄ device for monitoring the working status and vibrations of mechanical components in high-temperature environments.”

Figure S22 (R15). The output signals of the (HaaOH)BF₄ piezoelectric device at different temperatures.

Furthermore, we also conducted other measurements on ferroelectricity and piezoelectricity at high temperatures, such as temperature-dependent domain structure evolution (Figure S14), domain revolution (Figure S16), polarization switching (Figure S12) and piezoresponse measurements (Figure S17). These results indicate that the (HaaOH)BF₄ can still maintain ferroelectricity and piezoelectricity at high temperatures.

3. Although the authors have demonstrated by PFM that (HaaOH)BF₄ remains ferroelectric at 393 K, there is still a long way to go before its phase transition temperature (>528 K). Please add the variable-temperature electric hysteresis loop and variable-temperature d_{33} before the thermal decomposition temperature to increase the convincing data at the experimental level.

Authors' response: The decomposition temperature of (HaaOH)BF₄, determined through both differential scanning calorimetry (DSC) measurement and

thermogravimetric analysis, is found to be 528 K, below which no structural phase transition occurs. Therefore, we raised the temperature as high as possible to conduct the variable-temperature polarization switching and d_{33} measurements. Firstly, we characterized the local polarization switching behavior on the (HaaOH)BF₄ thin film at various temperatures through temperature-dependent SSPFM measurement. Evidently, the PFM phase curves exhibited square-shaped hysteresis loops with 180° contrast under the electric field, while the PFM amplitude curves displayed the typical butterfly shape at different temperatures. These curves clearly demonstrate the switching and hysteresis behaviors of ferroelectric polarization in (HaaOH)BF₄ during the off-field period. As the temperature increased, we observed a decrease in the voltage required for polarization switching (Figure R16). And the polarization could still be switched when the temperature reached 473 K (Figure R16). However, SSPFM curves at higher temperatures could not be obtained due to the impact of thermal disturbances generated by temperature rise on the test signals. Nonetheless, the temperature-dependent SSPFM results can already prove the ferroelectricity and piezoelectricity of (HaaOH)BF₄ at high temperatures. The additional experiment results have been added to the Supplementary Information (Figure S12), and the analysis and description have been supplemented in the revised manuscript as: “Meanwhile, temperature-dependent SSPFM measurements indicate that the required voltages for polarization switching decrease as the temperature increases and the polarization switching still occurs at 473 K (Figure S12).”

Figure S12 (R16). Temperature-dependent switching spectroscopy PFM measurements. The PFM phase loops and the PFM amplitude curves at (a) 323 K, (b) 353 K, (c) 383 K, (d) 413 K, (e) 443 K and (f) 473 K. The PFM phase curves are represented in purple, and the PFM amplitude curves are represented in green.

Furthermore, the temperature-dependent piezoelectric coefficient (d_{33}) was measured on the as-grown single-crystal of (HaaOH)BF₄ through the quasi-static method. The d_{33} coefficient remained stable at around 22 pC/N along the proximity of the [001] direction of the crystal until approximately 515 K (Figure R17a). This indicates that (HaaOH)BF₄ can maintain the noncentrosymmetric structure and piezoelectricity until the decomposition temperature. Meanwhile, we used the piezoresponse force microscopy (PFM) technique to measure the temperature-dependent local piezoresponse of (HaaOH)BF₄ films. The acquired curves exhibited typical resonance peaks which fitted well by the simple harmonic oscillator model. The effective amplitude was determined through dividing the amplitude by the quality factor. It was observed that (HaaOH)BF₄ films still exhibited significant piezoresponse up to 513 K (Figure R17b). This is consistent with the result of the temperature-dependent d_{33} measurement. The additional experiment results have been added to the Supplementary Information

(Figure S23), and the analysis and description have been supplemented in the revised manuscript as: “This is consistent with the temperature-dependent piezoresponse measurements, which indicate that the sample can maintain piezoelectricity until T_d (Figure S23).”

Figure S23 (R17). Temperature-dependent piezoelectric properties of (HaaOH)BF₄. (a) Piezoelectric coefficient (d_{33}) of (HaaOH)BF₄ at different temperatures. (b) PFM resonance peaks at different temperatures.

4. The authors tested that (Haa)BF₄ does not have a piezoelectric effect ($d_{33} = 0$), yet there is still a clear electrical signal response in Fig. 5(c), which is contradictory, please check the experimental data.

Authors' response: The comment is significant in piezoelectric devices and will interfere with experimental results. To perform control experiments with (HaaOH)BF₄ devices, we remade polydimethylsiloxane (PDMS) blank devices and (Haa)BF₄ devices using the same methods and measured the output voltage under the identical testing condition. As shown in Figure R18, both the PDMS blank device and the (Haa)BF₄ device exhibit significantly lower electrical signal response compared to (HaaOH)BF₄. This demonstrates that (HaaOH)BF₄ has superior output performance due to its piezoelectric property. However, the PDMS and (Haa)BF₄ devices also exhibit low output voltage about 0.08 V. This could be attributed to the device's architecture which leads to triboelectric generation signals between PDMS, electrodes, and the sample

when pressure is applied. The low electrical signal response in control devices also proves that the output of (HaaOH)BF₄ devices is generated by piezoelectricity. We have updated the data in Figure 5c and modified the description in the manuscript as: “Under the same condition, the blank PDMS device and (Haa)BF₄ device were tested, whose V_{OC} were close to 0 and were lower than (HaaOH)BF₄ device (Figure 5c). This indicates that the V_{OC} of the (HaaOH)BF₄ device is induced by piezoelectricity.”

Figure R18. Generated V_{OC} of energy-harvesting devices based on (HaaOH)BF₄ and (Haa)BF₄ polycrystalline samples under a periodical vertical force of 17 N at a frequency of 10 Hz, and the V_{OC} of the device without samples under the same conditions.

5. More detailed experimental details and test instrument models need to be given: including hysteresis loop, dielectric constant, piezoelectric coefficient tests, and energy harvesting applications.

Authors' response: The comments are meaningful and helpful for us to refine and improve the content of the manuscript. For the experimental details and test instrument

models, we have added detailed descriptions of hysteresis loop, dielectric constant, piezoelectric coefficient tests and energy harvesting applications in the Methods of the revised manuscript. We also added a Figure R19 to clearly depict the device structure and system configuration of piezoelectric measurement. The description was added to the manuscript as: **“Ferroelectric hysteresis loop.** The double-wave method was utilized to test the polarization-voltage curve on a homemade measuring system composed of a voltage source (Trek 609E-6), waveform generator (Keysight 33500B) and current meter (Keithley 6517B). Testing probes with indium gallium alloy were connected to different positions on the sample film, respectively, for collecting the in-plane current signals when triangular-wave voltage was applied.

Dielectric measurements. The samples were made with pressed-powder pellet and single crystals oriented perpendicular to the corresponding crystal axis, respectively. Silver conductive paste deposited on the plate surfaces was used as electrodes. Complex dielectric permittivities were measured with a TH2828A impedance analyzer over the frequency range from 500 Hz to 1 MHz with an applied electric field of 0.5 V.

Piezoelectric coefficient measurements. For macroscopic piezoelectric coefficient (d_{33}), we adopted the quasi-static method by a commercial piezometer (Piezotest, model: PM200). The single crystal plates for (Haa)BF₄ and (HaaOH)BF₄ were clamped between two flat metal plates, then a dynamic force of about 0.25 N along different axes was applied.

Preparation and measurements of piezoelectric energy-harvesting devices. All device fabrication processes were performed under ambient air and room-temperature conditions. The piezoelectric energy-harvesting devices consist of the top conductive adhesive tape, (HaaOH)BF₄, and the bottom conductive adhesive tape. The entire assembly is encapsulated using polydimethylsiloxane (PDMS) and cured at room temperature for one day. The formed piezoelectric devices were polarized under a direct current voltage. The preparation process for (Haa)BF₄ and blank devices is identical as well.

To measure the output performance, a fabricated device was placed on a custom-design mechanical system, where the device was periodically pressed by a shaker (active area

2×2 cm²). The force applied on the device was instantly monitored by a force sensor (502F01, YMC) during the measurement. A digital oscilloscope (DSO4032A, Keysight) was used to record the output performance of devices (Figure S25). To measure the sensor performance, a device was secured onto a dummy model. After grounding the upper electrode and connecting the upper and bottom electrodes to a digital oscilloscope (SDS6034H10Pro, SIGLENT), the output performance was tested by tapping and assessing its response.”

Figure S25 (R19). Device structure and testing system configuration. (a) Device structure of (HaaOH)BF₄ and sensor testing circuit. (b) Testing system configuration for piezoelectric devices. (c) Testing system configuration for lighting up the LEDs with mechanical pressure.

6. Does d_{33} vary along different crystal axis directions? The authors are suggested to give the relevant experimental data.

Authors' response: The reviewer's suggestion regarding the investigation of the piezoelectric coefficients of the crystal is crucial. We conducted quasi-static method to

test the piezoelectric coefficients along the three crystal axis directions of both (Haa)BF₄ and (HaaOH)BF₄ separately. As shown in Figure R20, (Haa)BF₄ exhibits no piezoelectric response along the three axes, and the piezoelectric coefficients for (HaaOH)BF₄ along the *a*- and *b*-axes are also 0 pC N⁻¹. However, only (HaaOH)BF₄ showed $d_{33} = \pm 22$ pC N⁻¹ along the *c*-axis. This aligns with the polarization direction of (HaaOH)BF₄ along the *c*-axis, and the piezoelectric coefficients are measured as 0 pC N⁻¹ along the other two axes perpendicular to the polarization direction. This indicates that the piezoelectric coefficients vary along different crystal axis directions, with the most pronounced piezoelectric coefficients observed along the polarization direction. The relevant experimental data were supplemented in the Supplementary Information (Figure S18). The analysis and description were added to the manuscript as: “On the contrary, (Haa)BF₄ has no piezoelectric response along different crystal axis directions as expected (Figure S18). The piezoelectric coefficient d_{33} along the corresponding polarization direction of (HaaOH)BF₄ is 22 pC N⁻¹, according to the quasi-static method (Berlincourt method) (Figure 5a).”

Figure S18 (R20). Piezoelectric coefficient d_{33} on (HaaOH)BF₄ (a) and (Haa)BF₄ (b) crystals using the quasi-static method along different axis directions.

7. The coordinate numbers in some of the figures are so small that they are not readable, so please standardize the format.

Authors' response: The comments are meaningful and helpful for us to improve the manuscript in relation to the format of the figures. We have enlarged the figures and the coordinate number in Figure 2-6 and Figure S11-S16 for ease of viewing and reading. And corresponding modifications have been made in the manuscript and Supplementary Information.

Reviewer #3 (Remarks to the Author):

This manuscript mentions the application of hydrogen bond modification design strategy to introduce spontaneous polarization and increase the phase transition temperature in molecular materials. The introduction of hydroxyl group on (Haa)BF₄ enabled the formation of intermolecular hydrogen bond network and achieved the synthesis of a new molecular ferroelectric (HaaOH)BF₄. Meanwhile, the intermolecular hydrogen bond can stable the polarization in molecular ferroelectric for a longer period of time. Based on (HaaOH)BF₄, the manuscript also explores the production of piezoelectric energy-harvesting devices and the application in self-powered sensing. The hydrogen bond modification mentioned in the manuscript is a very effective strategy for introducing polarization and optimizing performance in molecular materials. And the new molecular ferroelectric (HaaOH)BF₄ also has good properties and device performance. The crystal engineering strategy and the molecular ferroelectric provide new opportunities and platforms for the development of modern energy and micro-nano electronic devices. Therefore, I think this manuscript is suitable for publication in Nature Communications with few revisions and improvements.

1. (Haa)BF₄ undergoes a structural phase transition at 192 K, and the crystal structure will be different from room temperature. Thus, it is necessary to supplement the crystal structure of (Haa)BF₄ at low temperature and add some analysis and discussion.

Authors' response: The reviewer made a very good suggestion, and we have carried out the temperature-dependent single-crystal X-ray diffraction to measure the structure of (Haa)BF₄ at 163 K. The crystal of (Haa)BF₄ underwent a phase transition at 192 K and the structure was different from room temperature. The crystal structure of (Haa)BF₄ crystallizes in the space group *P*2₁/*c* (point group *2/m*) in the low-temperature phase, and both anions and organic cations are in the ordered state (Figure R21a). The (Haa)⁺ organic cations have a slight rotational shift (Figure R21b), and the N...F distance is also closer compared with the high-temperature phase structure (Figure R21c). The arrangement of organic cations and inorganic anions can be seen more clearly through

the unit cell stacking diagram (Figure R21d). Furthermore, the Hirshfeld surface analysis reveals a decline in the proportion of H⋯F contacts relative to the high-temperature phase, concomitant with a rise in the ratio of H⋯H contacts (Figure R21e). This is attributed to the ordered arrangement of N-H and B-F in the low-temperature phase, resulting in a reduction of the actual ratio of contacts simulated by the Hirshfeld surface analysis. The relevant results were supplemented in the Supplementary Information (Figure S8). The analysis and description were added to the manuscript as: “The temperature-dependent single-crystal X-ray diffraction shows that the low-temperature phase of (Haa)BF₄ crystallizes in the space group *P*2₁/*c* (point group *2/m*). And the distance between atoms and the intermolecular force have changed a little from the high-temperature phase due to the ordering of organic cations and anions and structural phase transition (Figure S8).”

Figure S8 (R21). Crystal structure of (Haa)BF₄ at 163 K. (a) Asymmetric units of (Haa)BF₄. (b) Packing view along the *c*-axis. The anions and cations are ordered. Parts of hydrogen atoms are omitted for clarity. (c) Hirshfeld surfaces of the guest cations in (Haa)BF₄. (d) Packing view of crystal structures of (Haa)BF₄. (e) Decomposed

fingerprint plots and proportion for the H···O, H···F and H···H contacts of the guest cations in (Haa)BF₄ on the Hirshfeld surfaces are displayed respectively.

2. The manuscript mentioned “the highest enhancement of phase transition temperature”. So, can the author add a statistical table showing the enhancement of phase transition temperature in molecular ferroelectrics? And the table may include compounds, different methods and the value of enhancement.

Authors’ response: The comments are meaningful and helpful for us to improve the manuscript in terms of content. We have compiled the relevant information on the enhancement of the phase transition temperature in molecular ferroelectrics, arranging them according to the degree of enhancement (Table R1). The table includes compounds before and after modification, phase transition temperatures, modification methods and the value of enhancement. In molecular ferroelectrics, strategies such as momentum matching, H/F substitution, deuterium isotope effect, steric confinement modulation and cage-confined ethylamine rotators have been employed to enhance the phase transition temperature. This work employed hydrogen bond modification to introduce polarization and enhance the phase transition temperature. The table and relevant references have been added in the supplementary information and the index has been added in the manuscript as: To our knowledge, the temperature enhancement has reached a reportedly record-high level in molecular ferroelectrics (Table S1).

Table S1 (R1). Summary of enhancing phase transition temperature in molecular ferroelectrics through various methods.

Prototype	T_1 (K)	Modified Compound	T_2 (K)	Modified Method	ΔT (K)	Ref.
(Haa)BF ₄	192	(HaaOH)BF ₄	528	Hydrogen Bond Modification	336	/
[(MeO–C ₆ H ₄ –NH ₃)(18-crown-6)][BF ₄]	127	[(MeO–C ₆ H ₄ –NH ₃)(18-crown-6)][TFSA]	415	Momentum Matching	288	¹
[ABCH]CdCl ₃	190	[4-FABCH]-CdCl ₃	419	H/F Substitution	229	²

(PD)PbI ₃	260	[4,4-DFPD] ₂ PbI ₄	429	H/F Substitution Steric	169	3
CM-iodized salt	324	CM-chloride salt	453	Confinement Modulation Steric	129	4
[H ₂ mdap]BiI ₅	264	[H ₂ mdap]BiCl ₅	377	Confinement Modulation	113	5
[N-MeDABCO]PbI ₃	363	[N-FMedabco]PbI ₃	473	H/F Substitution	110	3
(Pyrrolidinium)CdCl ₃	240	(R)- and (S)-3-F- (Pyrrolidinium)CdCl ₃	303	H/F substitution	63	6
[HDABCO][TFSA]	274	[DDABCO][TFSA]	327	Deuterium Isotope Effect Cage-Confined	53	7
(IBA) ₂ (EA)Pb ₂ Br ₇	326	(IBA) ₂ (EA) ₂ Pb ₃ Br ₁₀	370	Ethylamine Rotators	44	8
(Pyrrolidinium)MnCl ₃	295	(R)- and (S)-3- (Fluoropyrrolidinium) MnCl ₃	333	H/F Substitution	38	9
(Benzylammonium) ₂ Pb Br ₄	405	(Perfluorobenzylamm onium) ₂ PbBr ₄	440	H/F Substitution Steric	35	10
[Me ₃ NCH ₂ Cl]CdBrCl ₂	373	[Me ₃ NCH ₂ Cl]CdCl ₃	399	Confinement Modulation	26	11

MeO–C₆H₄–NH₃ = 4-Methoxyanilinium; TFSA = Bis(trifluoromethanesulfonyl)ammonium; ABCH = 1-Azabicyclo[2.2.1]heptane; 4-FABCH = 4-Fluoro-1-Azabicyclo[2.2.1]heptane; PD = Piperidinium; 4,4-DFPD = 4,4-Difluoropiperidinium; CM = Cyclohexylmethylammonium; H₂mdap = N-Methyl-1,3-Propanediamine; N-Medabco = Methyl-dabconium; N-FMedabco = N-Fluoromethyl-dabconium; DABCO = 1,4-Diazabicyclo[2.2.2] octane; IBA = Isobutylammonium; EA = Ethylammonium

3. There are obvious differences between the crystals of (HaaOH)BF₄ and (Haa)BF₄ in Figure S1. The crystal growth habits of (Haa)BF₄ can also be simulated by BFDH method and compared with (HaaOH)BF₄ and the actual crystal morphology.

Authors' response: We simulated the crystal growth habits of the (Haa)BF₄ at room temperature using the BFDH method. As shown in Figure R22-R23, the simulated crystal morphologies of (Haa)BF₄ and (HaaOH)BF₄ exhibit significant differences, yet both of them are consistent with the actual crystal morphologies, respectively (Figure R24). Furthermore, the crystal axes can be determined by the simulation (Figure R22-

R23). The dominant crystal face for (Haa)BF₄ is {200}, whereas for (HaaOH)BF₄ it is {110}. The difference may be attributed to variations in molecular structures and intermolecular forces between the two compounds. The simulated crystal structure of (Haa)BF₄ using the BFDH method was supplemented in the Supplementary Information (Figure S3). The analysis and description were added to the manuscript as: “The crystal morphologies of (Haa)BF₄ and (HaaOH)BF₄ (Figure S1) are the same as that predicted by the Bravais, Friedel, Donnay and Harker (BFDH) method (Figure S2-S3).”

hkl	Distance	Total facet area %
{1 1 0}	11.37673000	55.67300987
{0 1 1}	14.82842000	13.54068182
{0 1 -1}	14.82842000	13.54068182
{2 0 0}	15.28677999	4.46769528
{1 1 1}	16.68242939	6.14730970
{1 1 -1}	16.68242939	6.14730970

Figure S2 (R22). Morphology calculation of (HaaOH)BF₄ by BFDH.

hkl	Distance	Total facet area %
{2 0 0}	8.71679989	38.61050242
{1 0 1}	13.47964131	30.90171757
{2 1 0}	17.34295217	21.67539231
{0 1 1}	19.68503161	6.81739078
{1 1 1}	20.16174892	1.99499692

Figure S3 (R23). Morphology calculation of (Haa)BF₄ by BFDH.

Figure R24. The crystal photograph of (a) $(\text{Haa})\text{BF}_4$ and (b) $(\text{HaaOH})\text{BF}_4$.

4. Why the powder tablets of $(\text{Haa})\text{BF}_4$ are a little transparent, but the powder tablets of $(\text{HaaOH})\text{BF}_4$ is not?

Authors' response: $(\text{HaaOH})\text{BF}_4$ and $(\text{Haa})\text{BF}_4$ exhibit substantial differences in their crystal structures. It can be observed that both the organic cation and anion are ordered under the influence of hydrogen bonding in $(\text{HaaOH})\text{BF}_4$, while both of them are disordered in $(\text{Haa})\text{BF}_4$ (Figure R25). Plastic crystals are commonly found in compounds with spherical molecules, such as adamantane. (DOI: 10.1063/5.0039066) In the plastic crystal phase, the molecules undergo rapid rotator motions and their intermolecular interactions are weaker and more isotropic. The application of uniaxial pressure to microcrystalline powders causes a permanent deformation of each microcrystal without a fracture, resulting in a transparent state of the pressed pellets. Therefore, $(\text{Haa})\text{BF}_4$ due to the molecular disorder, exhibits plastic properties to some extent. However, its molecular rotational disorder is not entirely complete, so the pressed pellets only exhibit partial transparency. Nevertheless, the powder pellets of $(\text{Haa})\text{BF}_4$ and $(\text{HaaOH})\text{BF}_4$ still exhibit significant differences.

Figure S4 (R25). Pressed powder and crystal structure diagrams. (a) The schematic diagram of hydrogen bonds in (HaaOH)BF₄. (b) Pressed powders of (HaaOH)BF₄ (top) and (Haa)BF₄ (bottom). Packing view of crystal structures of (c) (HaaOH)BF₄ and (d) (Haa)BF₄ at 293 K.

5. Please add the process of making piezoelectric energy-harvesting devices and the sensing devices in Methods of the manuscript.

Authors' response: The comments are meaningful and helpful for us to improve the manuscript content. For the piezoelectric energy-harvesting devices and the sensing devices, we have added more detailed descriptions about experimental details and test instrument models in the Methods of the revised manuscript (Figure R26). We also added a Figure S25 to clearly depict the device structure and system configuration. The description was added to the manuscript as: **“Preparation and measurements of piezoelectric energy-harvesting devices.** All device fabrication processes were performed under ambient air and room-temperature conditions. The piezoelectric energy-harvesting devices consist of the top conductive adhesive tape, (HaaOH)BF₄, and the bottom conductive adhesive tape. The entire assembly is encapsulated using polydimethylsiloxane (PDMS) and cured at room temperature for one day. The formed piezoelectric devices were polarized under a direct current voltage. The preparation

process for (Haa)BF₄ and blank devices is identical as well.

To measure the output performance, a fabricated device was placed on a custom-design mechanical system, where the device was periodically pressed by a shaker (active area 2×2 cm²). The force applied on the device was instantly monitored by a force sensor (502F01, YMC) during the measurement. A digital oscilloscope (DSO4032A, Keysight) was used to record the output performance of devices (Figure S25). To measure the sensor performance, a device was secured onto a dummy model. After grounding the upper electrode and connecting the upper and bottom electrodes to a digital oscilloscope (SDS6034H10Pro, SIGLENT), the output performance was tested by tapping and assessing its response.”

Figure S25 (R26). Device structure and testing system configuration. (a) Device structure of (HaaOH)BF₄ and sensor testing circuit. (b) Testing system configuration for piezoelectric devices. (c) Testing system configuration for lighting up the LEDs with mechanical pressure.

6. Response time is a very important parameter in piezoelectric sensors. So, what is the

value of the response time of the piezoelectric sensors based on the molecular ferroelectric?

Authors' response: The comment is significant in piezoelectric sensors and will display the fundamental properties. We collected and amplified a signal from a heavy tapping. And we obtained the response time of the (HaaOH)BF₄ piezoelectric sensor device in touch mode by marking its rise range. As shown in Figure R27, the pressure sensor responds to the pressure quickly, with a fast response time of 10.24 ms. The response time of (HaaOH)BF₄ piezoelectric sensors was supplemented in the Supplementary Information (Figure S26). The analysis and description were added to the manuscript as: "Meanwhile, the piezoelectric sensor responded to the pressure quickly, with a fast response time of 10.24 ms under a pressure upon finger tapping (Figure S26)."

Figure S26 (R27). The piezoelectric sensor shows a response time of 10.24 ms under tapping with the finger.

We would like to express our greatest appreciation to the reviewer for his/her recognition of our academic research, and also for the comments and suggestions to help us improve the quality of this manuscript as well as our future research indeed. We have made appropriate modifications to improve the scholarly quality. And we will

conduct more in-depth research of hydrogen bond modification in molecular ferroelectrics.

REVIEWERS' COMMENTS

Reviewer #1 (Remarks to the Author):

They have answered my questions.

Reviewer #2 (Remarks to the Author):

The authors have addressed all my concerns and I recommend publishing the paper in Nature Communications.

Reviewer #3 (Remarks to the Author):

The revised manuscript can be accepted in the present state.

Author's response

(Authors' response was colored in blue and quote of addition/revision of manuscript was colored in purple.)

Reviewer #1 (Remarks to the Author):

They have answered my questions.

Reviewer #2 (Remarks to the Author):

The authors have addressed all my concerns and I recommend publishing the paper in Nature Communications.

Reviewer #3 (Remarks to the Author):

The revised manuscript can be accepted in the present state.

Authors' response to all reviewers:

First of all, we would like to express our greatest appreciation to all reviewers for their affirmation of the academic innovations in our manuscript. Moreover, we are grateful to reviewers for their questions and suggestions, which help us to improve the quality of this manuscript as well as our future research indeed. We hope that the improvements can convince the reviewers that our paper is suitable to be published in Nature Communications.